# Zero-to-Hero: Enhancing Zero-Shot Novel View Synthesis via Attention Map Filtering

**Ido Sobol**[1]    **Chenfeng Xu**[2]    **Or Litany**[1,3]

[1]**Technion**    [2]**UC Berkeley**    [3]**NVIDIA**

https://zero2hero-nvs.github.io/

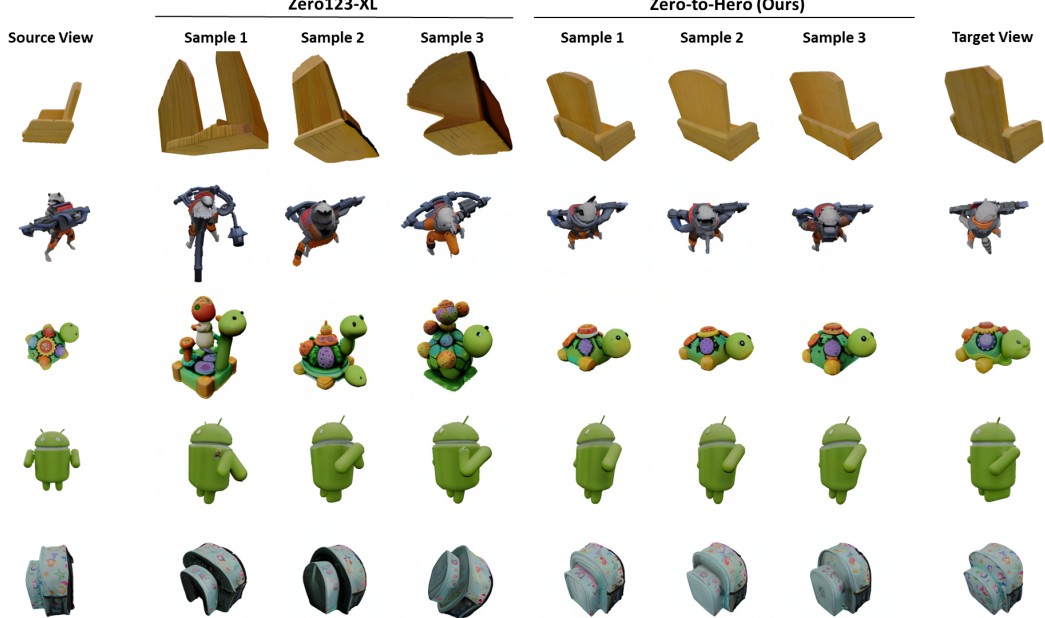

Figure 1: Novel views generated from a single source image (far left column) at a specific target view angle (with different seeds), compared between Zero123-XL [27] and our Zero-to-Hero method. Operating during inference, our method achieves significantly higher fidelity and maintains authenticity to the original image, all while ensuring realistic variation in the results (e.g. variations in chair backs in the top row). The ground-truth target view is displayed in the far right column.

## Abstract

Generating realistic images from arbitrary views based on a single source image remains a significant challenge in computer vision, with broad applications ranging from e-commerce to immersive virtual experiences. Recent advancements in diffusion models, particularly the Zero-1-to-3 model, have been widely adopted for generating plausible views, videos, and 3D models. However, these models still struggle with inconsistencies and implausibility in new views generation, especially for challenging changes in viewpoint. In this work, we propose Zero-to-Hero, a novel test-time approach that enhances view synthesis by manipulating attention maps during the denoising process of Zero-1-to-3. By drawing an analogy between the denoising process and stochastic gradient descent (SGD), we implement a filtering mechanism that aggregates attention maps, enhancing generation reliability

38th Conference on Neural Information Processing Systems (NeurIPS 2024).

and authenticity. This process improves geometric consistency without requiring retraining or significant computational resources. Additionally, we modify the self-attention mechanism to integrate information from the source view, reducing shape distortions. These processes are further supported by a specialized sampling schedule. Experimental results demonstrate substantial improvements in fidelity and consistency, validated on a diverse set of out-of-distribution objects. Additionally, we demonstrate the general applicability and effectiveness of Zero-to-Hero in multi-view, and image generation conditioned on semantic maps and pose.

# 1   Introduction

The pursuit of realistic image synthesis at arbitrary views, given only a single source image, has long been a cornerstone challenge in computer vision and graphics. This technology can cater to countless applications, such as interactive product inspection, robot-scene interaction, and immersive virtual experiences. In this work, we aim to advance this important line of research by improving the generation of novel views that are plausible and faithful to the input image. A recent, promising approach, Zero-1-to-3 [27] has developed a foundation model to synthesize novel views based on a single source image and a target view angle. By leveraging a pre-trained, image-conditioned stable diffusion model backbone [3], fine-tuned with target camera poses, and trained on paired source and target views from a vast collection of 3D models [10, 9], Zero-1-to-3 can generalize beyond its training set and generate plausible novel views. As a result, this model has quickly gained popularity, inspiring subsequent work in 3D and 4D scene generation [8, 24, 39, 28, 26, 25, 57, 33, 45, 35, 20].

Despite its remarkable ability, Zero-1-to-3 has been observed to generate views that are implausible, or inconsistent with the input object in terms of shape and appearance [24, 8]. Previous works have tried to mitigate these issues by retraining diffusion models with more data [9] or to generate multiple views [39, 25, 26, 24, 8, 28, 56]. Despite substantial improvement, both approaches are resource-intensive due to the required re-training on large-scale 3D datasets. Another line of work attempts to consolidate inconsistencies across multiple generated views through a 3D representation like NeRF [14, 30]. However, direct aggregation often results in blurry outputs, as observed by [24]. Instead, ViVid-1-to-3 [24] employed a multiview representation that naturally supports the use of a video foundation model. Nevertheless, this approach requires generating the entire trajectory from the source to the target view, adding significant complexity and computational overhead. Notably, the denoising process in Zero-1-to-3 remains unchanged.

In this work, we propose Zero-to-Hero, a novel test-time technique that addresses view synthesis artifacts through attention map manipulation. Recognizing attention maps as crucial for latent predictions, we hypothesize that enhancing robustness in attention maps predictions can significantly reduce generation misalignment. To achieve this, we draw an analogy between the denoising process in diffusion models and stochastic gradient descent (SGD) optimization of neural networks. Specifically, we relate network weights that predict local gradients at each optimization step, based on sampled training examples and labels, to the denoising network's attention maps that predict latent representations from sampled noise at each denoising step. In this work, we view the generation (denoising) process as an unrolled optimization, with attention maps as parameters of a score prediction model. Inspired by gradient aggregation and weight-averaging techniques that improve prediction robustness (e.g., consistency training [47]), we propose a filtering mechanism to enhance attention map reliability. This mechanism comprises iterative map aggregation within denoising steps and map averaging across denoising steps. The result is more reliable maps, particularly during the early denoising stages when coarse output shapes are determined, leading to more plausible and realistic views.

To further promote consistency with the input, we modify the self-attention operation by running a parallel generation branch using the identity pose, incorporating its keys and values into the attention layer of the target view. Unlike previous applications of this technique [29, 5, 1]), we find it beneficial in view synthesis to limit its use to the early denoising stages, preventing shape distortions. Our unique denoising procedure is further complemented by a novel sampling schedule that emphasizes early and late denoising stages, maximizing performance. Our main contributions are as follows:

- To address the main limitations of the Zero-1-to-3 model, we perform an in-depth analysis and identify self-attention maps as the main candidate for correcting generation artifacts.

- We establish a conceptual analogy between model weights in stochastic gradient descent-based network training and the role of attentions map updates during generation of a denoising diffusion model. Based on this, we propose a simple yet powerful attention map filtering process resulting in enhanced target shape generation. We supplement our filtering technique with identity view information injection and a specialized sampling schedule.

- Our method requires no additional training, and it avoids the overhead of external models or generating multiple views.

Through comprehensive experiments on out-of-distribution objects, we demonstrate that our technique robustifies Zero-1-to-3 and its extended version, Zero123-XL, leading to views that are more faithful to both the input image and desired camera transformation. Our results show significant and consistent improvements across both appearance and shape evaluation metrics. Additionally, we find that Zero-to-Hero naturally generalizes to additional tasks including multi-view, and image generation conditioned on semantic maps and pose. In all cases, we observed significant improvement in condition following and visual quality.

## 2 Related Work

### 2.1 Novel View Synthesis with Diffusion Models

Diffusion models have dominated various generative applications [17, 11, 38, 36, 41]. Particularly, novel-view synthesis, as a core of applications like augmented reality and simulations, naturally enjoys the benefits of high-fidelity zero-shot synthesis with diffusion models. One line of works [51, 27, 44, 22, 54, 24] is to generate a novel-view image given a source image (*i.e.,* image-to-image). These approaches typically involve training a diffusion model conditioned on both an arbitrary camera pose and the source view. For instance, the representative work, Zero-1-to-3, fine-tunes a pre-trained Stable Diffusion model [36] by replacing the text prompt with camera pose and CLIP features[34]. Moreover, another research trajectory [32, 21, 14, 7, 46] proposes generating a 3D representation from a single image (*i.e.*, image-to-3D), which allows for sampling desired views from these 3D models. Our method, Zero-to-Hero, builds on the first approach (specifically Zero-1-to-3 and Zero123-XL) and distinguishes itself by eliminating the need for extensive training. Instead, it offers a test-time, plug-and-play approach that significantly enhances visual quality and consistency.

### 2.2 Test-Time Refinement in Diffusion-Based Generation

A common test-time refinement strategy in diffusion generation is leveraging guidance [29, 2, 18] to direct the sampling process with additional conditions. For example, Repaint [29] utilizes a mask-then-renoise strategy to refine the generation results. Repaint also introduces a per-step resampling technique, where given a horizon-size h, a latent $z_t$ is re-noised to $z_{t+h}$ and then denoised again to $z_t$ multiple times. They observe that resampling helps to generate more harmonized outputs, given an external guidance or condition. Restart [55] offers a sampling algorithm based on a variation of resampling within a chosen interval of steps. Our method is inspired by the strategy of per-step resampling. We show that it serves as a powerful correction mechanism throughout the generation process, even when no external guidance or condition is provided.

### 2.3 Attention Map Manipulation in Diffusion Models

Stable Diffusion [36] utilizes attention to enforce the condition information onto the generated results. Previous works demonstrate that manipulating the attention operation can achieve new capabilities [49, 5, 1, 57]. For example, MasaCtrl [5] uses Mutual Self-Attention where source and target images are generated jointly while sharing information, by injecting source image keys and values to the target through self-attention. Here we employ Mutual Self-Attention in the context of novel view synthesis. Differently to prior works we find it beneficial to limit its use to the early denoising stages.

## 3 Background

### 3.1 Zero-1-to-3: Challenges and Limitations

Zero-1-to-3 is a pioneering method for novel view synthesis based on a diffusion model, which has gained significant popularity. This model is built upon the image-conditioned variant of Stable

Diffusion (SD) [36], fine-tuned specifically for novel view synthesis. Zero-1-to-3 is conditioned on a source image and relative transformation to the desired view angle $[\mathcal{R}|\mathcal{T}]$. Maintaining the SD architecture, these conditions are integrated in two ways. First, a CLIP [34] embedding of the input image is concatenated with the relative transformation $[\mathcal{R}|\mathcal{T}]$ and mapped to the original CLIP dimension to form a global pose-CLIP embedding, interacting with the UNet layers through cross-attention, enriching the generation with high-level semantic information. In parallel, the input image is channel-concatenated with the denoised image, helping the model preserve the identity and details of the synthesized object.

While Zero-1-to-3 [27] has achieved substantial progress in novel view synthesis, several common issues limit its practical application. Firstly, the generated images might not fit real-world distributions, resulting in implausible and unrealistic outputs (e.g., first row in Fig.1). Secondly, the target image may appear plausible but be inconsistent with the input image in terms of shape or appearance (e.g., fifth row in Fig.1).

In this work, we identify the critical role of self-attention maps in high-quality generation and propose a novel filtering process that enhances robustness without requiring further training. This process addresses the aforementioned issues, ensuring reliable and consistent results.

## 3.2  Leveraging Gradient and Weight Aggregation for Improved Prediction Consistency

In this work, we draw a conceptual analogy between gradients and weights in stochastic gradient descent (SGD), and latents and attention maps in denoising diffusion models. Leveraging this analogy, we adapt techniques from SGD to enhance prediction consistency in diffusion models. Below, we summarize general techniques in SGD that improve the training process.

SGD is a fundamental tool in network training [4], designed to navigate the weight (network parameter) space towards local minima. For a neural network $F(x; \theta)$ with parameters $\theta$, SGD samples training data points $x_i$ and their corresponding labels $y_i$, and computes the gradient of the loss function $L(F(x_i; \theta), y_i)$ with respect to $\theta$ to update the parameters. In practice, aggregation of *gradients* and network *weights* during training is often performed to reduce variance and improve convergence. Gradient aggregation typically involves averaging gradient values over a batch, while weight aggregation accounts for the history of the weights in each update.

Notable examples include temporal averaging in Adam optimizer [23], Stochastic Weight Averaging (SWA) [19] and teacher networks [47] used in consistency training by employing an exponential moving average (EMA) of a student network to maintain high-quality predictions. This technique is prevalent in semi-supervised and representation learning [13, 15, 6]. For a detailed study of EMA in network training, we refer readers to [31].

## 4  Method

In this work we are concerned with the task of single image novel view synthesis. Formally, given an input image of an object and a relative camera transformation towards a desired target view, our goal is to generate the image at that target view. Specifically, we build upon the seminal work of Zero-1-to-3 [27] which tackled this task via a diffusion model. As detailed in Sec. 3.1 Zero-1-to-3 often struggles to generate plausible and input-consistent images. In this work we propose Zero-to-Hero– a novel training-free technique that significantly improves its generation quality through attention map manipulation. This section is structured as follows. (4.1) through network architecture analysis we identify self-attention maps as key for robust view generation; (4.2) drawing inspiration from SGD convergence-enhancement techniques, we outline our novel attention map filtering pipeline; (4.3) details each step of the map filtering; (4.4) introduces the mutual self-attention which we use for shape guidance at early generation stages; (4.5) Finally, our proposed hourglass scheduler is introduced for more efficient utilization of generation steps. Fig. 2 depicts Zero-to-Hero's main modules. An ablation of these modules is provided in Tab. 2 and in Sec. 8.6 of the appendix.

### 4.1  Analyzing the Role of Cross- and Self-Attention Layers in Novel View Generation

Zero-1-to-3 builds upon the image-to-image variant of Stable Diffusion [36], which utilizes a UNet [37] architecture as its backbone and incorporates both self- and cross-attention layers. In this subsection, we analyze the roles of these components and their contributions to the generated view. This analysis aims to identify effective intervention points for enhancing generation quality.

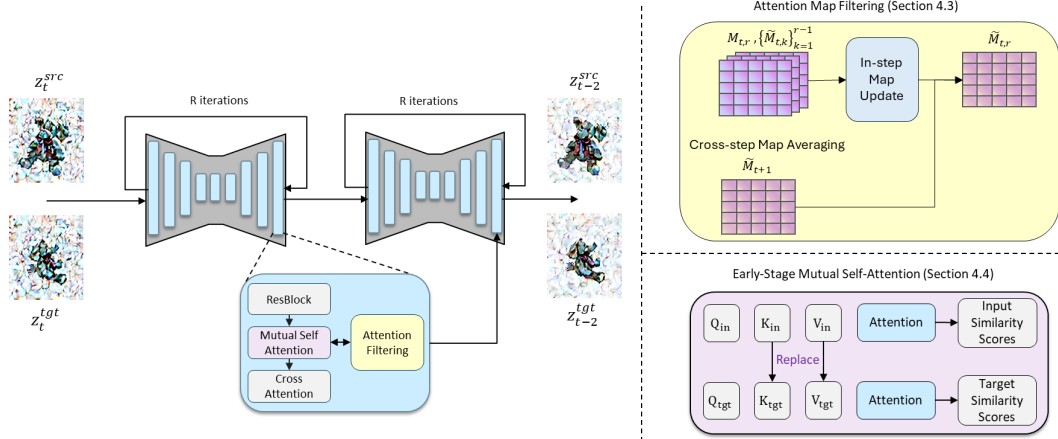

Figure 2: **Zero-to-Hero main modules.** (Left) Two denoising steps of the generation process of both the source (top) and target views (bottom). Each denoising step is iterated $R$ times ("resampling"). (Right-top) **Attention map filtering**: Robustifying attention maps via an aggregation of same step and previous steps attention maps. (Right-bottom) **Mutual self-attention**: Guiding target shape through the keys and values of the source generation branch.

**Global pose conditioning through cross-attention.** Prior work [16] has demonstrated that the cross-attention layers in text-to-image diffusion models, which link text tokens to the latent image, are spatially aware and can be used for spatial manipulation. We first investigate these cross-attention layers, as they are the only components in the model through which the target pose is injected. In the text-to-image variant of Stable Diffusion, the generation is conditioned on a prompt $\mathcal{P}$ containing $C$ tokens, each encoded with CLIP into an embedding, resulting in a condition $c_{T2I} \in \mathbb{R}^{C \times d_{CLIP}}$. However, in Zero-1-to-3, the condition is a pose-CLIP embedding $c \in \mathbb{R}^{1 \times d_{CLIP}}$, project to keys $K_t \in \mathbb{R}^{1 \times d}$ Formally, given a sample $z_t \in \mathbb{R}^{H \times W \times d_z}$ and its corresponding queries $Q_t \in \mathbb{R}^{H \times W \times d}$, the pre-softmax attention map $\mathcal{A}$ between $Q_t$ and $K_t$ has dimensions $H \times W \times 1$. Given that the summation of the softmax is always 1, the post-softmax attention map $softmax(\mathcal{A})$ in Zero-1-to-3 is a constant all-ones matrix. A visual demonstration is presented in Fig. 7 in the appendix.

The post-softmax attention map is used to compute a weighted sum over the values matrix $V \in \mathbb{R}^{1 \times d}$, obtained by a transformation of the condition $c$. Since the attention matrix is an all-ones matrix, *we conclude that the cross-attention operation degenerates into a global bias term, lacking spatially aware operations*. Computing similarity scores in the cross-attention layers is redundant as these scores are never used. While in principle it is possible to improve the global bias term by additional optimization objectives and extra training overhead, we focus on the self-attention layers to enhance the results and mitigate the consistency issues while avoiding retraining the model.

**Spatial information flow through self-attention.** By monitoring the self-attention layers during the generation process, we observe that random noise introduced to the latent representation also introduces randomness to the attention maps. This randomness, while promoting generation diversity, can often lead to undesired strong correlations, that are misaligned with the true target. These strong correlations may persist through the denoising process, resulting in accumulated errors and visual artifacts.

Given the above observation and the insight about the spatial-degeneracy in the cross-attention layers, we hypothesize that the self-attention layers preserve the information about the structure and geometry of the generated image, through the similarity scores between different elements in the latent vector. To validate our hypothesis, we conduct a straightforward experiment to assess Zero-1-to-3's performance using the 'ground truth' self-attention maps, which reflect the most accurate connections considering the true target image. Specifically, we selected two images, $I^{src}$ and $I^{tgt}$, with known relative camera parameters $[\mathcal{R}|\mathcal{T}]$. We first encode $I^{tgt}$ to obtain the clean latent $z_0^{tgt}$ and then add subtle noise to obtain the corresponding noisy latent for timestep $\tau_{init}$, $z_{init}^{tgt}$. A single denoising step is performed on the noisy latent, and we save the self-attention maps from each layer in the UNet,

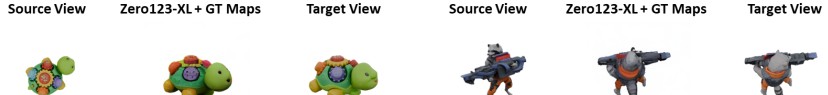

Figure 3: Through the injection of ground-truth attention maps extracted from the target view, we demonstrate that **Self-attention maps are key to robust view synthesis**.

considering these maps as the ground truth (GT) maps. In our experiments, $\tau_{\text{init}} = 5$. Next, we sample random Gaussian noise and denoise it to regenerate the target image. During each denoising step, we replace the computed self-attention maps with the GT maps, without altering any other components (e.g., cross-attention layers, residual blocks) or the latents, queries, keys, and values. We report the results in the experiment section in Tab. 1, showing a significant improvement in all metrics. Note that the results obtained with the GT attention maps are a strict upper bound, as the GT maps contain information about areas that are invisible in the source view. Our experiment validates that through improved self-attention maps, the generated image becomes more plausible. Visual examples are shown in Fig. 3 and in Fig. 8 in the appendix, refer to Fig. 1 for the results of Zero123-XL and Zero-to-Hero for the same views.

## 4.2    From SGD to Diffusion Models: Attention Map Filtering as Weight-Space Manipulation

We draw a conceptual analogy between a denoising process of a diffusion model, and SGD based network optimization as both progress through gradient prediction of a loss function, and log probability (the score)[43], respectively. Building on the discussion from the previous section, we treat the self-attention maps as parameters $M$ in the denoising process $z_{t-1} = \mu(z_t; M_t, \psi)$ and define their update process as a mapping[1]: $M_t \to z_{t-1} \to M_{t-1}$. Here the map $M_{t-1}$ results from passing the latent $z_{t-1}$ through the attention layers. This process is analogous to gradient descent optimization in neural networks, where each step adjusts the weights in the direction of the gradient of a loss function, such as the log probability in classification tasks. We denote this weight update as a mapping $\theta_t \to \hat{y}_t \to \theta_{t+1}$, where $\hat{y}_t = F(x_t; \theta)$, and the updated parameters $\theta_{t+1}$ result from a gradient step.

As detailed in Sec. 3.2, gradient and weight aggregation are essential for robust convergence. We outline this process in three steps illustrating the analogy between network parameter updates and attention map filtering. Fig. 4 further illustrates the analogy.

---

**Step-by-Step Analogy**  From network parameters to attention maps

---

**Network Training**

1: **Sampling**: Generate $R$ pairs of samples and their corresponding labels $\{(x_i, y_i)\}_{i=1}^{R}$.
2: **In-step weight update**: Average the gradients to adjust the network parameters: $\tilde{\theta}_t = \theta_t - \lambda \sum \nabla_\theta L(F(x_i; \theta_t), y_i)$.
3: **Cross-step weight averaging**: Update network parameters using EMA: $\theta_{t+1} = \alpha\theta_t + (1-\alpha)\tilde{\theta}_t$, for $\alpha \in [0, 1]$.

**Denoising Process**

1: **(Re-)Sampling**: Repeatedly re-noise $z_{t-1}$ to $z_t$ and denoise, $R$ times.
2: **In-step map update**: Resmapling results in a sequence of maps $\{M_{t,r}\}_{r=1}^{R}$, aggregated into the final updated prediction $\widetilde{M}_{t,R}$.
3: **Cross-step map averaging**: Update attention maps via EMA: $\widetilde{M}_{t-1} = \alpha M_t + (1-\alpha)\widetilde{M}_{t-1}$, for $\alpha \in [0, 1]$.

---

## 4.3    Robust View Generation via Attention Map Filtering

We now discuss in detail each of the map filtering steps. A scheme of the different filtering modules is provided in Fig. 2 (Left, and Top-right).

**Latent refinement via per-step resampling.** Inspired by previous studies [29, 2], we implement per-step resampling throughout the image generation process. We select a range of timesteps

---

[1]Although the latent prediction $z_{t-1}$ depends on the parameters $\psi$ of denoising UNet $\mu$, they remain unchanged. Here we emphasize that the attention maps $M$ are the parameters being updated.

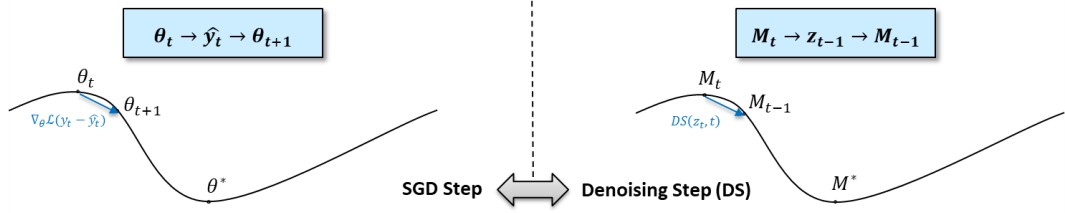

Figure 4: From SGD to Diffusion Models: An illustration of our conceptual analogy.

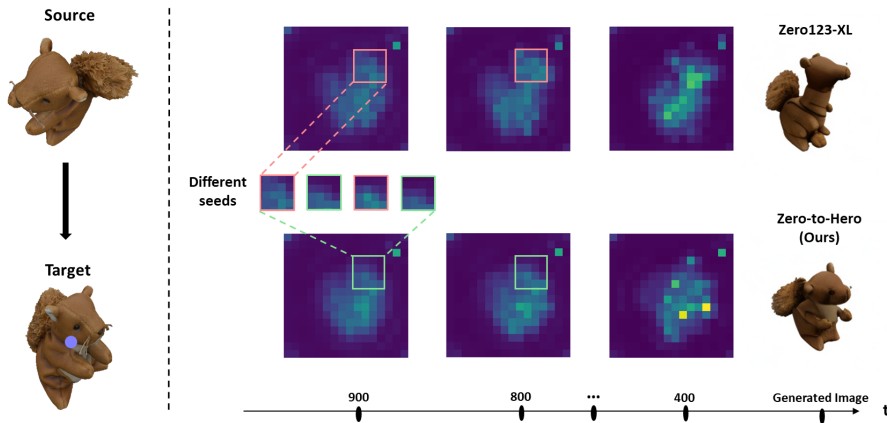

Figure 5: **Attention map filtering in action.** We compare the attention scores of zero123-XL (top) and Zero-to-Hero (bottom) wrt the region marked with a purple circle at different denoising steps. Both methods are initialized with the same seed. We observe that the strong correlation values in the upper right corner lead to exaggerated content creation (note the unrealistically elongated neck). Conversely, through filtering, Zero-to-Hero mitigates these artifacts, leading to robust view synthesis.

$[t_{min}, t_{max}]$, where each denoised sample $z_t$ is re-noised with the proper noise ratio to the previous sampled timestep $z_{t+1}$ and denoised back to $z_t$ for $R$ iterations[2]. We concur with previous studies that resampling functions as an effective corrective mechanism to the generated image, as can be seen in Tab. 2). Experimentally, we find that it is particularly useful during the *initial stages* of the denoising process. Through resampling we progressively generate $R$ attention maps with different noise patterns. We propose to leverage these intermediate maps to further boost performance through in- and cross-step attention map manipulations.

**In-step map update.** We propose a novel attention pooling function $f$, to update the attention maps within the denoising step. Specifically, a self-attention map is refined based on previous maps $\{M_{t,k}\}_{k=1}^{r-1}$ created at the same timestep. Since resampling is a sequential process, we perform a progressive aggregation scheme: $\widetilde{M}_{t,r} = f(M_{t,r}, \{\widetilde{M}_{t,k}\}_{k=1}^{r-1})$. We found the element-wise min-pooling operator $f(a, b) = min(a, b)$ to perform best in our experiments. We discuss other options for $f$ in the appendix.

**Cross-step map averaging.** Resampling tends to favour "conservative" generations, often gradually neglecting fine image details like buttons and eyes as denoising progresses. This phenomenon it not resolved by the in-step update. To mitigate this issue, we propose to pass the self-attention map at time $t$, $\widetilde{M}_t$ to the next step in the denoising process. We implement this cross-step aggregation via EMA: $\widetilde{M}_{t-1} = \alpha M_t + (1 - \alpha)\widetilde{M}_{t-1}$, for $\alpha \in [0, 1]$. This method effectively balances past priors with current data to enhance the denoising results. In practice we apply both methods in tandem, as detailed in Sec. 8.5. An example of our attention filtering is presented in Fig. 5. Note that our filtering mechanism is applied to the pre-softmax attention maps.

---

[2]Depending on the sampling algorithm, $h \geq 1$ steps may be taken.

### 4.4 Early-Stage Shape Guidance via Mutual Self-Attention

Of the two challenges describe in the introduction, attention map filtering handles well generating realistic outputs. Yet, we observe it is sometimes insufficient to enforce consistency with the input appearance and structure. We propose to utilize mutual self-attention as a complementary technique, to propagate information from the input to the target. Similar to prior works [49, 5, 1, 57, 52], we generate the input and target views in parallel. At each timestep $t$, we obtain queries, keys and values of the self-attention layers for the input and the target branches. Similar to the mutual self-attention introduced in [5], we modify the self-attention operation of the target by replacing the target keys and values with those of the input branch: $Attn(Q^{tgt}, K^{in}, V^{in})$ (Fig. 2 (bottom-right)).

Previous studies initiate mutual self-attention (MSA) at a later denoising step (small $t$). We find that at that stage, the structure has already been determined. Instead, to guide the structure of the target to be more consistent with the input, we find it more effective when applied from the beginning of the denoising process. While MSA is effective at transferring the appearance and structure of the input to the target, it may overfit. We find that *terminating the MSA process once the structure has been stabilized becomes crucial* in mitigating overfitting. In practice we find that applying MSA during the first third of the denoising process is a good rule of thumb for optimal results. We refer the reader to Sec. 8.5 in the appendix for further details.

We view early-stage MSA as a complementary filtering scheme. Building on the property that generating the input view (which the model is conditioned on) is a much easier task for the model compared to generating novel views, keys and values predicted in the input branch are "cleaner". Mapping them to the target view thus refines the predicted scores, facilitating a more stable and reliable output.

### 4.5 Hourglass Sampling Scheduler

As resampling is a time-consuming operation, significantly increasing the number of function evaluation (NFE), we seek an efficient scheduling scheme that will enable robust and high quality generation, while preserving the fast generation times of Zero-1-to-3. We use the common DDIM sampling [42], and propose an efficient scheduling scheme to select the sampled timesteps. In the experiments section, we demonstrate that increasing the number of denoising steps does not necessarily improve the performance, and therefore we aim to use an overall small number of sampled timesteps. Specifically, we find that denser sampling during the beginning of the denoising process is crucial to promote realism. We also find that denser sampling at the final steps enhances fine details. Therefore, we suggest a double heavy-tailed scheduling scheme we call *Hourglass*, according to which we divide the generation process into 3 stages, as detailed in 8.5. Within each stage, we sample steps uniformly via DDIM. However, in the first and last stages we sample steps more densely (at a higher rate) than in the middle stage, by a factor of $\lambda_{den}$.

## 5 Experiments

**Datasets.** We evaluate our method on two datasets, following previous works. Firstly, *Google Scanned Objects (GSO)* Dataset [12], which includes 1030 scanned household objects. However, the dataset's imbalance (e.g., 254 objects are categorized as "shoe") and the high proportion of symmetrical shapes limit its reliability for evaluation. To address this, inspired by [24], we select a challenging subset of 50 objects from GSO, avoiding symmetrical and repetitive shapes. For each object, we render 8 random views (details in Sec. 8.9 of the appendix) and synthesize the remaining views from each source view, generating each target view with 3 different seeds to calculate the average score per measure. Secondly, *RTMV* Dataset [48], which consists of 3D scenes. Each scene contains 20 random objects. For evaluation, we randomly select 50 scenes, and 8 random views per scene. The evaluation process is the same as described for GSO.

**Metrics.** We report the image quality metrics PSNR, SSIM [50] and LPIPS [59]. As these metrics are sensitive to slight color variations, we segment the generated targets and their corresponding real images, via thresholding, and report the Intersection Over Union (IoU) score.

### 5.1 Evaluations

**Quantitative evaluation.** We evaluate Zero-to-Hero against zero-1-to-3 and on zero123-XL. We report all metrics for the original models using 25, 50 and 100 DDIM steps, and for our method applied to both models. In Tab. 1 and in Tab. 3, we provide the results for GSO and RTMV datasets,

respectively. We include the number of sampled timesteps T and the total number of network evaluation NFE (accounting for resampling). Zero-to-Hero consistently improves performance across all metrics, taking a significant step towards bridging the performance gap to GT attention maps. All implementation details, including analysis of the inference cost of our modules, are provided in Sec. 8.5 of the appendix.

Table 1: **Quantitative evaluation on a challenging GSO subset.** Zero-to-Hero consistently improves performance upon baselines, taking a significant step towards oracle map performance (bottom rows).

| Name | T | NFE | PSNR ↑ | SSIM ↑ | LPIPS ↓ | IOU ↑ |
|---|---|---|---|---|---|---|
| Zero-1-to-3 | 25 | 25 | 17.27 | 0.851 | 0.173 | 73.5% |
| Zero-1-to-3 | 50 | 50 | 17.24 | 0.850 | 0.173 | 73.5% |
| Zero-1-to-3 | 100 | 100 | 17.21 | 0.850 | 0.173 | 73.4% |
| Ours (Zero-1-to-3) | 26 | 66 | **17.67** | **0.859** | **0.163** | **75.3%** |
| Zero123-XL | 25 | 25 | 17.72 | 0.854 | 0.163 | 76.4% |
| Zero123-XL | 50 | 50 | 17.71 | 0.854 | 0.163 | 76.4% |
| Zero123-XL | 100 | 100 | 17.68 | 0.854 | 0.163 | 76.4% |
| Ours (Zero123-XL) | 26 | 66 | **18.35** | **0.864** | **0.153** | **78.3%** |
| Zero-1-to-3 + GT maps | 50 | 50 | 21.52 | 0.888 | 0.122 | 88.8% |
| Zero123-XL + GT maps | 50 | 50 | 21.79 | 0.890 | 0.117 | 88.6% |

**Qualitative evaluation.** In Fig. 1, Fig. 9 and Fig. 10, we visually demonstrate how Zero-to-Hero is able to mitigate various artifacts generated by Zero-1-to-3, from implausible results to incorrect poses. In Fig. 1 and in Fig. 9, we present 3 examples per target view, generated with 3 random seeds, to emphasize the consistency and robustness our method offers.

## 5.2 Ablation Study

To assess the contribution of Zero-to-Hero different components to the final performance, we evaluate our pipeline on our challenging GSO subset, starting from the baseline Zero123-XL and gradually adding each module. The results are summarized in Tab. 2. When reporting the results of the vanilla Zero123-XL, we use its *best* score achieved by running the model with 25, 50, and 100 steps. A similar comparison against zero-1-to-3 is included in Sec. 8.6 of the appendix, demonstrating consistent behavior. We also analyze the affect of resampling and our attention filtering on the *generation diversity* in Sec. 8.7 of the appendix. Additionally, we provide qualitative results, demonstrating the common contributions of AMF and MSA in Fig. 10 in the appendix.

Table 2: **Ablation Study.** We demonstrate the importance of each of Zero-to-Hero modules, applied to the base method Zero123-XL: Sample scheduling (Hourglass), Resampling (Resample), Attention map filtering (AMF), and Early-Stage Mutual Self-Attention (MSA). Consistent conclusions are reached with the base model Zero-1-to-3 and are shown in Sec. 8.6 of the appendix.

| Hourglass | Resample | AMF | MSA | PSNR ↑ | SSIM ↑ | LPIPS ↓ | IOU ↑ |
|---|---|---|---|---|---|---|---|
| ✗ | ✗ | ✗ | ✗ | 17.72 | 0.854 | 0.163 | 76.4% |
| ✓ | ✗ | ✗ | ✗ | 17.74 | 0.855 | 0.162 | 76.6% |
| ✓ | ✓ | ✗ | ✗ | 17.85 | 0.857 | 0.160 | 77.1% |
| ✓ | ✓ | ✓ | ✗ | 17.92 | 0.859 | 0.157 | 77.8% |
| ✓ | ✓ | ✗ | ✓ | 18.25 | 0.862 | 0.155 | 77.6% |
| ✓ | ✓ | ✓ | ✓ | **18.35** | **0.864** | **0.153** | **78.3%** |

## 5.3 Attention Map Filtering Beyond Novel View Synthesis

Although our work addresses the core limitations of single view synthesis models, the condition enforcing effect of our Attention Map Filtering (AMF) is more general. We have conducted several preliminary experiments which demonstrate promising results. Further details are provided in Sec. 8.8.

**Conditional image generation**. A brief study of ControlNet models [58] demonstrated that they suffer from similar limitations as Zero-1-to-3 and its follow ups. Namely, lack of condition enforcement and frequent appearance of visual artifacts. We implemented our proposed AMF module for two pre-trained ControlNet models. We provide qualitative results for pose- and segmentation-conditioned ControlNet models in Fig. 6 and Fig. 13, respectively.

**Multi-view synthesis**. We integrate AMF into MVDream [40], a text-to-multiview model, and find that it helps to mitigate the same issues as in the single view case. In Fig. 14, we provide qualitative results.

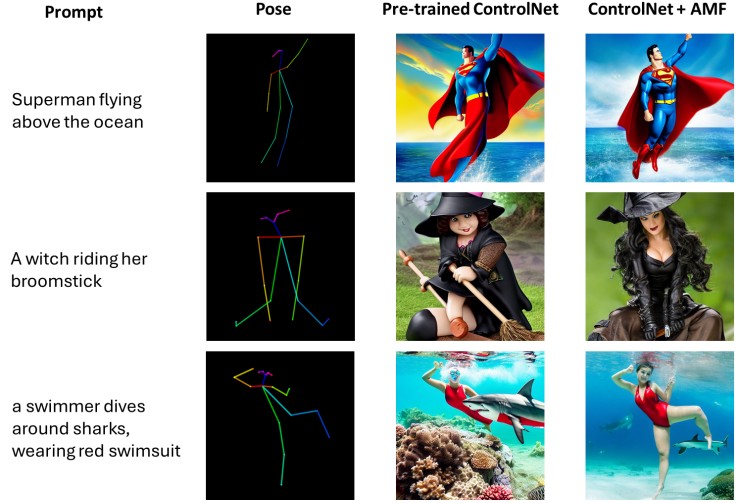

Figure 6: **Qualitative results for pose-conditioned ControlNet.** Qualitative results for pre-trained pose-conditioned ControlNet, without and with AMF. Both methods are initialized with the same seed. AMF leads to results that are more plausible and better align with the conditions.

## 6 Conclusions and Future Work

In this paper, we introduced Zero-to-Hero, a training-free method to boost the robustness of novel view synthesis. We enhanced the performance of a pre-trained Zero-1-to-3 diffusion model using two key innovations: a test-time attention map filtering mechanism that enhances output realism, and an effective use of source view information to improve input consistency. Our approach also features a novel timestep scheduler for maintaining computational efficiency. In future work, we aim to refine our method by developing trainable filtering mechanisms, enhancing pose authenticity via cross-attention manipulation, and extending our approach to other diffusion-based generative tasks.

**Limitations.** Our method, operating at test-time, is limited by the generative capabilities of the pre-trained model. If Zero-1-to-3 cannot correctly generate the target pose, our method may not enhance the output, as demonstrated in the inset figure.

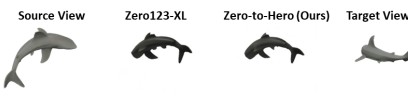

**Broader impact.** Through enhancing view synthesis, our method offers significant benefits to applications in virtual reality, augmented reality, and robotics. As it requires no further training, it is readily accessible. However, this accessibility also simplifies the creation of realistic novel views, which could be exploited for malicious purposes such as deepfakes.

## 7 Acknowledgments

We sincerely thank Matan Atzmon for impactful discussions and James Lucas for his invaluable feedback. Or Litany is a Taub fellow and is supported by the Azrieli Foundation Early Career Faculty Fellowship. This research was supported in part by an academic gift from Meta. The authors gratefully acknowledge this support.

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

# 8 Appendix

## 8.1 Cross-Attention in Zero-1-to-3

We demonstrate the degenerated cross-attention layers in Zero-1-to-3 in Fig. 7. We generate a random target view using Zero123-XL and extract the cross-attention map generated at timestep 900 in the last layer of the UNet. The same behaviour holds across all timesteps and layers. The displayed map is created by averaging over all the attention heads.

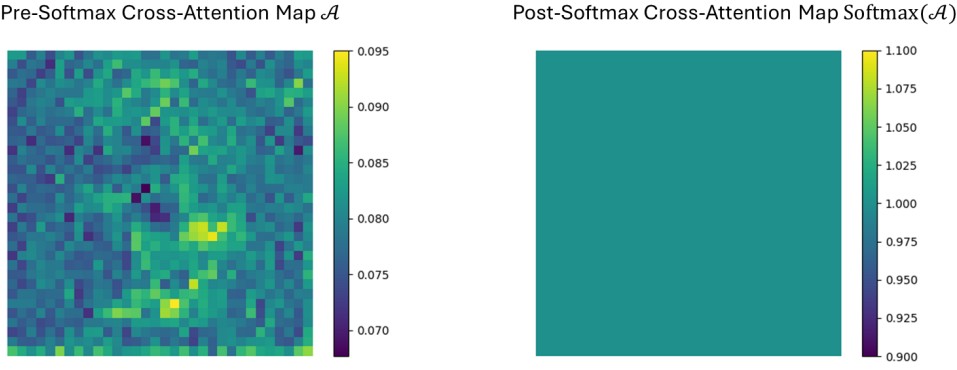

Figure 7: **Cross-Attention in Zero-1-to-3.** (Left) The cross-attention map before applying softmax. (Right) The degenerated all-ones attention map, produced by applying softmax on the left map.

## 8.2 Self-Attention in Zero-1-to-3

In Fig. 8 we present additional examples showing the performance of Zero123-XL when the self-attention maps are replaced with the 'ground truth' maps, extracted from the real target as described in Sec. 4.1.

## 8.3 Qualitative Results

In Fig. 9, we provide more qualitative results that reinforce the effectiveness of Zero-to-Hero.

## 8.4 Quantitative Results

On the GSO evaluation, shown in Tab. 1 in the main paper, Zero123-XL improved performance over Zero-1-to-3 by utilizing 12x more data, yielding gains of [0.45, 0.003, -0.01, 2.9%] in PSNR, SSIM, LPIPS, and IoU, respectively. When applied to Zero-1-to-3, Our method achieved comparable gains, [0.40, 0.008, -0.01, 1.8%], without using any additional data or further training. Notably, when applied to Zero123-XL, our method resulted in even larger improvements [0.63, 0.1, -0.01, 1.9%] (over Zero123-XL), demonstrating that these performance boosts cannot be solely achieved with merely more data.

Additionally, in Tab. 3, we report all metrics on RTMV dataset, showing substantial improvement over baselines.

## 8.5 Implementation Details

We evaluated our method on the default checkpoint of Zero-1-to-3 and on Zero123-XL. We used the same hyper-parameters for both models, as follows. All experiments were run on a single NVIDIA RTX 4090.

Note that the length of the forward process of Stable Diffusion is $T = 1000$.

**Attention map filtering pipeline.** As mentioned in the method section of the main paper, we apply both in-step and cross-step aggregation in tandem. In detail, we preserve attention map history

| Source View | Zero123-XL + GT Maps | Target View | Source View | Zero123-XL + GT Maps | Target View |

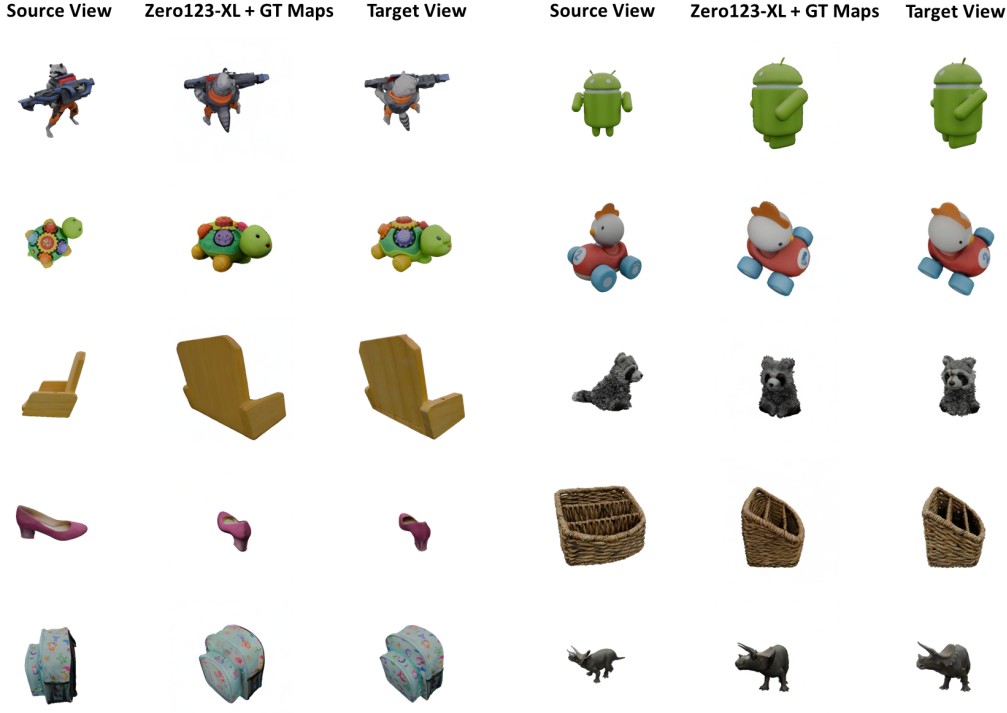

Figure 8: Through the injection of ground-truth attention maps extracted from the target view, we demonstrate that **self-attention maps are key to robust view synthesis**.

Table 3: **Quantitative evaluation on RTMV dataset.** Zero-to-Hero consistently improves performance upon baselines.

| Name | T | NFE | PSNR ↑ | SSIM ↑ | LPIPS ↓ | IOU ↑ |
|---|---|---|---|---|---|---|
| Zero-1-to-3 | 25 | 25 | 10.27 | 0.587 | 0.396 | 70.7% |
| Zero-1-to-3 | 50 | 50 | 10.26 | 0.585 | 0.395 | 70.7% |
| Zero-1-to-3 | 100 | 100 | 10.25 | 0.584 | 0.394 | 70.7% |
| Ours (Zero-1-to-3) | 26 | 66 | **10.86** | **0.617** | **0.373** | **71.2%** |
| Zero123-XL | 25 | 25 | 10.64 | 0.595 | 0.391 | 71.9% |
| Zero123-XL | 50 | 50 | 10.61 | 0.593 | 0.391 | 72.0% |
| Zero123-XL | 100 | 100 | 10.64 | 0.593 | 0.390 | 72.1% |
| Ours (Zero123-XL) | 26 | 66 | **11.15** | **0.618** | **0.372** | **72.8%** |

information denoted as $H_t$. Subsequently, we integrate this historical information through a simple blending technique at each denoising step, along with our in-step map update:

$$\widetilde{M}_{t,r} = \alpha_c \cdot f(M_{t,r}, \{\widetilde{M}_{t,k}\}_{k=1}^{r-1}) + (1 - \alpha_c) \cdot H_{t+1},$$

where $\alpha_c \in [0, 1]$, $f$ is an attention pooling function and $t + 1$ represents the previous sampled step.

The historical information is updated at the last resampling iteration (the R-th iteration) of each timestep as follows: $H_t = \alpha_h \cdot \widetilde{M}_{t,R} + (1 - \alpha_h) \cdot H_{t+1}$, where $\alpha_h$ is a decay factor within the range $[0, 1]$. This historical data is initialized from the first refined self-attention map at the final resampling step, expressed as $H_T = \widetilde{M}_{T,R}$.

**Resampling** is performed during timesteps $t \in [800, 1000]$, with R=5.

**In-step map update.** In-step map update is applied at timesteps t $\in [800, 1000]$, with element-wise min-pooling as a denoising function in all our experiments. We find that min-pooling is useful during the earlier steps of the denoising, where the noise is substantial.

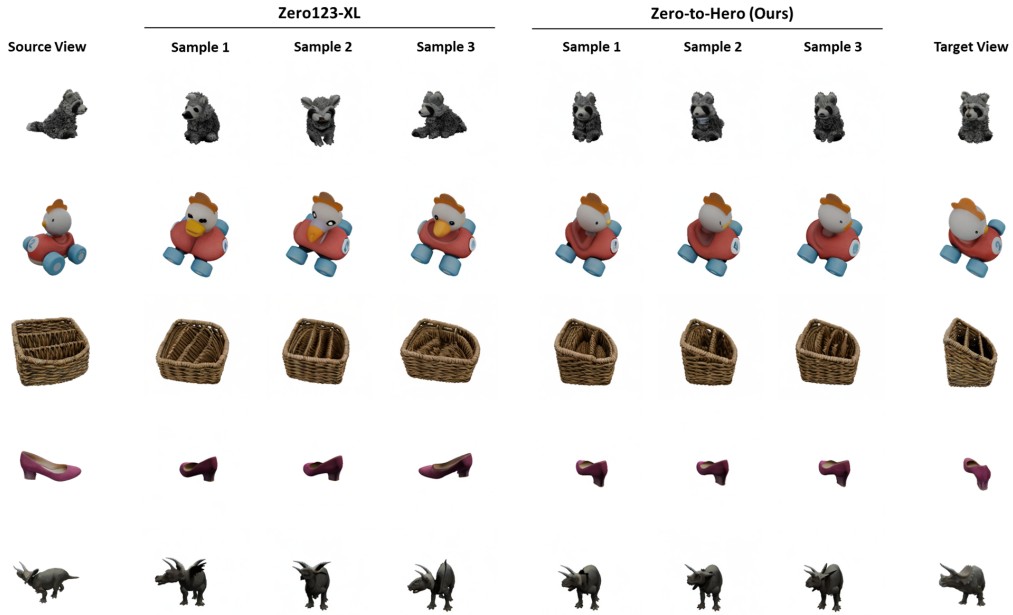

| | Zero123-XL | | | Zero-to-Hero (Ours) | | | |
| Source View | Sample 1 | Sample 2 | Sample 3 | Sample 1 | Sample 2 | Sample 3 | Target View |

Figure 9: Novel views generated from a single source image (far left column) at a specific target view angle (with different seeds), compared between Zero123-XL [27] and our Zero-to-Hero method. The ground-truth target view is displayed in the far right column.

We explored several simple denoising functions:

1. Averaging: $\widetilde{M}_{t,r} = \frac{1}{r}(M_{t,r} + \sum_{k=1}^{r-1} \widetilde{M}_{t,k})$, which can be extended to weighted averaging.

2. Min pooling: $\widetilde{M}_{t,r} = min(M_{t,r}, \{\widetilde{M}_{t,k}\}_{k=1}^{r-1})$.

3. Exponential Moving Average: $\widetilde{M}_{t,r} = \alpha_i \cdot M_{t,r} + (1 - \alpha_i) \cdot \widetilde{M}_{t,r-1}$, for $\alpha_i \in [0,1]$.

While all three improved performance over vanilla resampling, we find min-pooling to work best during the early stages of the denoising process.

**Cross-step map averaging.** We perform cross-step information passing at timesteps $t \in [600, 1000]$, with $\alpha_h = 0.5$ and $\alpha_c = 0.2$.

**Early-stage mutual self-attention.** We apply Early-Stage mutual self-attention on timesteps t $\in[600, 1000]$ for Zero123-XL and on timesteps t $\in[700, 1000]$ for Zero-1-to-3. As mentioned in Sec. 3, the input latent vector is concatenated to the generated latent vector in Zero-1-to-3. Therefore, the target queries contain information about the input view, introducing a bias towards the input view during the generation process. Although mutual self-attention manages to enforce consistency with the input, it sometimes causes pose shifts and visual artifacts. Therefore, We only apply mutual self-attention in the decoder layers of the UNet with spatial resolution of $16 \times 16$ and $32 \times 32$ (the two largest resolutions, since the latent dimension of Zero-1-to-3 is $32 \times 32$).

**Hourglass scheduling.** We divide the denoising process into 3 generation stages. We define the early stage during timesteps $t \in [\tau_e, \tau_T]$, where the general shape and geometry are determined. $\tau_T$ is the length of the forward process[17, 42]. We define the middle stage for $t \in [\tau_m, \tau_e]$, where both the shape and appearance are refined gradually. We define the last stage for $t \in [0, \tau_m]$, where mostly fine-details are refined and added. As describes in the main paper, within each stage we sample steps uniformlly via DDIM sampling. However, the sampling rate is higher for the early and late stages.

In our experiments, we determine the different stages of the denoising process as follows, setting $\lambda_{den} = 5$. The early stage is defined for $t \in [800, 1000]$ and we sample 10 timesteps uniformly within this interval. The middle stage is defined for $t \in [200, 800]$ and we sample 6 timesteps uniformly

within this interval. The last stage is defined for $t \in [0, 200]$ and we sample 10 timesteps uniformly within this interval. In total, we sample 26 timesteps throughout the denoising process.

**Setting the hype-parameters.** All hyperparameters were tuned based on a small validation set of objects. Due to limited computational resources, the tuning process was not exhaustive. We found that the method's performance is not highly sensitive to most parameters.

1. Cross-step weight (alpha): We experimented with values ranging from 0.1 to 0.9 in increments of 0.1. Alpha determines the weight assigned to previous predictions in the cross-step aggregation. In Zero123-XL, early predictions were generally reliable, so a larger weight yielded better results. We maintained the same parameters for Zero-1-to-3. In the experiments conducted with additional models such as ControlNet and MVDream, a smaller weight generally produced better outcomes.

2. Resampling iterations (R): We observed that the model's performance is relatively insensitive to the choice of R, with values between 4 and 8 yielding similar results. While some objects benefited from larger values ($\sim$10), the overall improvement was minimal. Additionally, as shown in Fig. 11, large values of R ($\sim$15-20) can reduce diversity. For additional experiments with models like ControlNet and MVDream, we fixed R at 5 and did not test other values.

3. Filtering schedule: We apply resampling and our filtering mechanism during timesteps $t \in [t_{min}, T]$. To set $t_{min}$, we consider values from 500 to 900, in increments of 100.

4. MSA schedule: We apply Early-stage MSA in timesteps $t \in [t_{min}, T]$. To set $t_{min}$, we consider values from 400 to 1000, in increments of 100.

**Additional Inference Cost Analysis.** Our proposed modules add a computational overhead to the base model. In the paper, we addressed this by counting the overall number of function evaluations (NFE) and keeping it on par with the base model to ensure a fair comparison. We discuss the individual computational overhead of each module.

1. Resampling: Similar to the total number of denoising steps, T, the number of resampling iterations, R, linearly increases the NFE. Our chosen value for R and the timesteps at which we apply it resulted in mapping 26 denoising steps to a total of 66 NFE. Resampling does not introduce additional computational cost over adding standard denoising steps.

2. Mutual self-attention: The main additional cost of MSA is the necessity to generate the input view in addition to the target views, meaning the effective number of generated samples is increased by one.

3. Attention map filtering: We apply AMF during the early steps of the denoising process. We find that the in- and cross-step operation adds a small computational overhead. We maintain two additional instances of each attention map: the attention map from the previous timestep (for cross-step updates) and the refined map from the current timestep (for in-step updates).

We provide running times (in seconds) of Zero-1-to-3 and Zero-to-Hero for the same NFE (66) in Tab. 4. Note that the number of samples reported in the table does not include the input view generated in Zero-to-Hero. Overall, we manage to provide competitive running times. If both MSA and AMF are active (requiring the generation of the input), the running time is increased by approximately 1-1.5 seconds. If only AMF is active, the overhead is much smaller, averaging around 0.5 seconds (there is no need to generate the extra input view).

Table 4: **Runtime analysis of the computational overhead of Zero-to-Hero.**

| Number of Samples | Zero-1-to-3 | Zero-to-Hero (Ours) |
|---|---|---|
| 1 | 2.2 | 3.1 |
| 2 | 2.9 | 3.9 |
| 3 | 3.5 | 4.9 |
| 4 | 4.3 | - |

## 8.6 Ablation Study

We conduct ablation studies focused on each component on our method, and present the results for the default checkpoint of Zero-1-to-3 and for Zero123-XL.

In addition to the main ablation table shown in the main paper, we present the same study on Zero-1-to-3 in Tab. 5. Additionally, we provide qualitative results, demonstrating the common contributions of AMF and MSA in Fig. 10.

**Hourglass scheduling.** We run the original model with 25, 50 and 100 steps sampled uniformly using DDIM, and show that the hourglass scheduling, with 26 steps sampled in total, performs better or on par. The main benefit of our Hourglass scheduling is reducing the number of steps, but it also slightly improves the overall performance. Results are presented for Zero-1-to-3 in Tab. 8 and for Zero123-XL in Tab. 6.

**Attention map filtering.** We use the hourglass scheduling for all further experiments. We run the original model with and without resampling and attention filtering. Our experiments demonstrate that adding the attention filtering in addition to the resampling mechanism improves the performance across all metrics. Results are presented for Zero-1-to-3 in Tab. 9 and for Zero123-XL in Tab. 7.

**Early-Stage mutual self-attention.** To the best of our knowledge, most prior works utilizing MSA fall into two categories:

1. Training-free based methods [5]: In these works, MSA is typically applied after the general structure of the target is formed to transfer appearance details from the input to the target. In this scenario, MSA does not contribute to the initial structure formation of the target.

2. Training or fine-tuning based methods [52, 53]: These works incorporate MSA layers within the training or fine-tuning process.

Our approach is distinct as we employ MSA in a training-free manner, but crucially, we apply it from the beginning of the denoising process until the target structure stabilizes—a phase we term "Early-Stage Shape Guidance". To validate the impact of early-stage MSA on structure, we conducted a simple experiment using Zero123-XL with 50 DDIM steps. We measure the effect of activating MSA at different stages of denoising on Intersection over Union (IoU) metric, noting that image quality metrics improved similarly. The results are reported in Tab. 10. Our approach demonstrate a significant improvement in the structural integrity of the image. Applying MSA in the later stages of the denoising process may hinder the results as it introduces a bias towards the input image.

Table 5: **Ablation study — Zero-1-to-3.** We demonstrate the importance of each of Zero-to-Hero modules, applied to the base method Zero-1-to-3: Sample scheduling (Hourglass), Resampling (Resample), Attention map filtering (AMF), and Early stage Mutual Self-Attention (MSA).

| Hourglass | Resample | AMF | MSA | PSNR ↑ | SSIM ↑ | LPIPS ↓ | IOU ↑ |
|---|---|---|---|---|---|---|---|
| ✗ | ✗ | ✗ | ✗ | 17.27 | 0.851 | 0.173 | 73.5% |
| ✓ | ✗ | ✗ | ✗ | 17.29 | 0.852 | 0.173 | 73.6% |
| ✓ | ✓ | ✗ | ✗ | 17.40 | 0.853 | 0.170 | 74.3% |
| ✓ | ✓ | ✓ | ✗ | 17.49 | 0.855 | 0.167 | 75.0% |
| ✓ | ✓ | ✗ | ✓ | 17.52 | 0.857 | 0.165 | 74.5% |
| ✓ | ✓ | ✓ | ✓ | **17.67** | **0.859** | **0.163** | **75.3%** |

Table 6: **Ablation study: Hourglass scheduling — Zero123-XL.** We demonstrate the superiority of our Hourglass scheduling over uniform DDIM sampling with different number of denoising steps. The experiments are based on Zero123-XL.

| Sampling | T | NFE | PSNR ↑ | SSIM ↑ | LPIPS ↓ | IOU ↑ |
|---|---|---|---|---|---|---|
| Uniform | 25 | 25 | 17.72 | 0.854 | 0.163 | 76.4% |
| Uniform | 50 | 50 | 17.71 | 0.854 | 0.163 | 76.4% |
| Uniform | 100 | 100 | 17.68 | 0.854 | 0.163 | 76.4% |
| Hourglass | 26 | 26 | **17.74** | **0.855** | **0.162** | **76.6%** |

Table 7: **Ablation study: Attention map filtering — Zero123-XL.** We demonstrate the importance of Attention Map Filtering (AMF) over only applying Resampling (Resample). The experiments are based on Zero123-XL.

| Name | T | NFE | PSNR ↑ | SSIM ↑ | LPIPS ↓ | IOU ↑ |
|---|---|---|---|---|---|---|
| Zero123-XL | 26 | 26 | 17.74 | 0.855 | 0.162 | 76.4% |
| + Resample | 26 | 66 | 17.85 | 0.857 | 0.160 | 77.1% |
| + Resample + AMF | 26 | 66 | **17.92** | **0.859** | **0.157** | **77.8**% |

Table 8: **Ablation study: Hourglass scheduling — Zero-1-to-3.** We demonstrate the superiority of our Hourglass scheduling over uniform DDIM sampling with different number of denoising steps. The experiments are based on Zero-1-to-3.

| Sampling | T | NFE | PSNR ↑ | SSIM ↑ | LPIPS ↓ | IOU ↑ |
|---|---|---|---|---|---|---|
| Uniform | 25 | 25 | 17.27 | 0.851 | 0.173 | 73.5% |
| Uniform | 50 | 50 | 17.24 | 0.850 | 0.173 | 73.5% |
| Uniform | 100 | 100 | 17.21 | 0.850 | 0.173 | 73.4% |
| Hourglass | 26 | 26 | **17.29** | **0.852** | 0.173 | **73.6**% |

Table 9: **Ablation study: Attention map filtering — Zero-1-to-3.** We demonstrate the importance of Attention Map Filtering over only applying Resampling (Resample). The experiments are based on Zero-1-to-3.

| Name | T | NFE | PSNR ↑ | SSIM ↑ | LPIPS ↓ | IOU ↑ |
|---|---|---|---|---|---|---|
| Zero123-XL | 26 | 26 | 17.29 | 0.852 | 0.173 | 73.6% |
| + Resample | 26 | 66 | 17.40 | 0.853 | 0.170 | 74.3% |
| + Resample + AMF | 26 | 66 | **17.49** | **0.853** | **0.167** | **75.0**% |

Table 10: **Ablation study: Attention map filtering — Zero-1-to-3.** We demonstrate the importance of Attention Map Filtering over only applying Resampling (Resample). The experiments are based on Zero-1-to-3.

| Method | Timesteps where MSA is applied | IOU ↑ |
|---|---|---|
| No MSA | - | 76.4% |
| All the way | 1000 to 0 | 76.6% |
| After structure is formed | 800 to 0 | 76.5% |
| Ours (from the beginning until structure is formed) | 1000 to 600 | **77.6**% |

## 8.7 Generation Diversity Ablation Study

While our attention filtering manages to generate more robust and consistent results, we find that excessive it might limit the generation diversity. We explore the two main factors that can reduce the diversity: the number of filtering iterations $R$ and the interval of timesteps $[t_{min}, t_{max}]$ where we apply filtering.

**Number of filtering iterations.** In Fig. 11, we show the effect of different values of $R$ on the diversity of the results. All other hyper-parameters remain unchanged. We find that as $R$ grows, the generation diversity reduces, while the level of realism improves

**Filtering schedule.** We apply filtering during timesteps $t \in [t_{min}, T]$. In Fig. 12, we show the effect of different values of $t_{min}$ on the diversity of the results. All the hyper-parameters besides $t_{min}$ remain unchanged. We observe that as we apply filtering for longer periods of the denoising steps, the results are more realistic but less diverse.

| Source View | Target View | Zero123-XL | + MSA | + AMF | Zero-to-Hero |
|---|---|---|---|---|---|

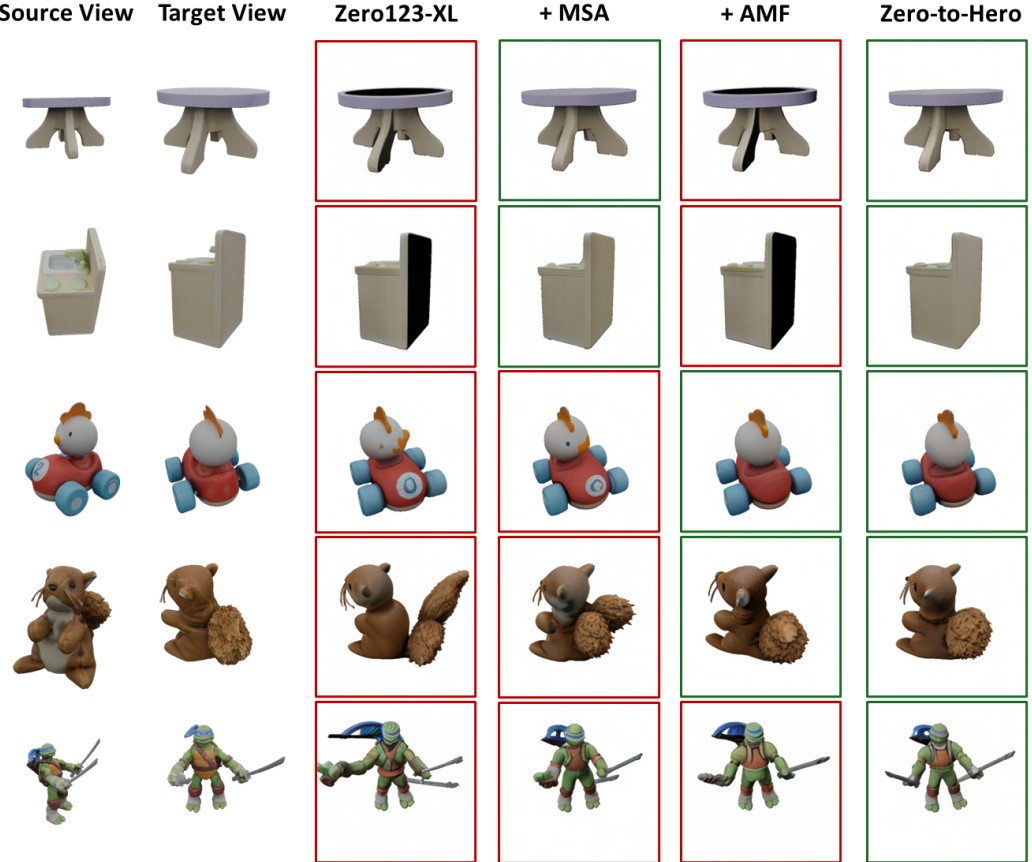

Figure 10: Qualitative results, illustrating where MSA and AMF excel. The first two rows present cases where MSA fixed black and random textures in the target views. This contribution shows a larger improvement in image quality metrics, although MSA usually cannot fully resolve significant structural issues. The third and fourth rows show structural improvements achieved with AMF. Finally, the last row shows a case where neither technique worked well enough on its own, but the combination did the work: the original result of Zero123-XL shows an implausible right hand and an incorrect pose of the right sword. MSA lead to more stable result, where the hand is not out-of-distribution, but could not resolve the incorrect pose of the sword. AMF lead to reasonable right hand and placement of the sword, but the sword is not aligned well the the texture of the original sword. The final result show correct structure and texture for the right hand and sword.

Through our ablation study, we find that in practice, a moderate application of attention maps filtering is sufficient for improving fidelity while preserving diversity. While excessive resampling might limit diversity (e.g. using large values of R), we only apply our filtering mechanism in the earlier steps of the denoising process, using a small value of R (5 in all our experiments). This enables our method to maintain diversity effectively.

When evaluating diversity against base models, we observed that some results of the base models were highly implausible, as can be seen in Fig 1 and Fig. 9 in the appendix. For example, Zero-1-to-3 produces out-of-distribution results, or results that are not aligned with either the input image or the target pose, leading to seemingly larger variance. However, this "diversity" is largely due to misalignment and artifacts, which comes at the expense of fidelity.

In our case, our model continues to generate diverse results that are both plausible and better aligned with the conditions (e.g. the various chair backs and turtles heights in Fig. 1).

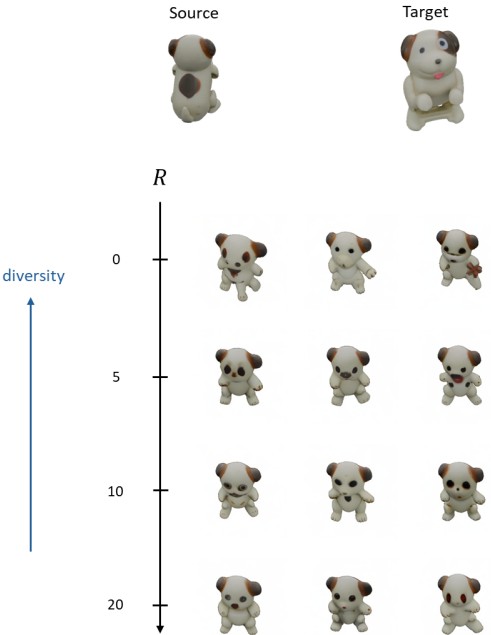

Figure 11: **Generation diversity wrt filtering iterations.** We select an extreme change in viewpoint, and show how different choices of $R$, the number of filtering iterations, affect the diversity of generated outputs. As filtering iterations increase, the results become less diverse and more realistic.

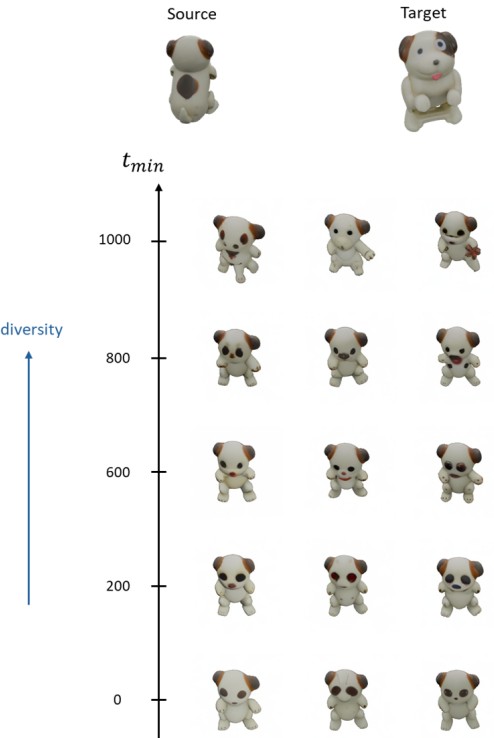

Figure 12: **Generation diversity wrt filtering schedule.** We select an extreme change in viewpoint, and show how different choices of the timesteps interval where filtering is applied, affect the diversity of results. As filtering is applied for longer periods during the denoising process, the results become less diverse and more realistic.

## 8.8 Attention Map Filtering Beyond Novel View Synthesis

In this subsection, we describe several preliminary experiments where we integrate attention map filtering to other base models. Remarkably, the integration of our proposed method into other base models is straightforward, demonstrating its applicability and simplicity.

In all the mentioned experiments, resampling is performed during timesteps $t \in [700, 1000]$, with R=5. Additionally, the cross-step parameters were slightly tuned for each base model. Mutual Self-Attention is not implemented, as it is not immediately applicable.

**Conditional image generation**. A brief study of ControlNet models [58] demonstrated that they suffer from similar limitations as zero123 and its follow ups. Namely, lack of condition enforcement and frequent appearance of visual artifacts. We implemented our proposed AMF module for two pre-trained ControlNet models. In Fig. 6 in the main paper, we provide qualitative results for pose-conditioned ControlNet. In Fig. 13, we provide qualitative results for segmentation-conditioned ControlNet. We find that attention map filtering robustly mitigates artifacts across various prompts and seeds for both base models.

**Multi-view synthesis**. We integrate AMF into MVDream [40], a text-to-multiview model, and find that it helps to mitigate the same issues as in the single view case. We provide qualitative results in Fig. 14.

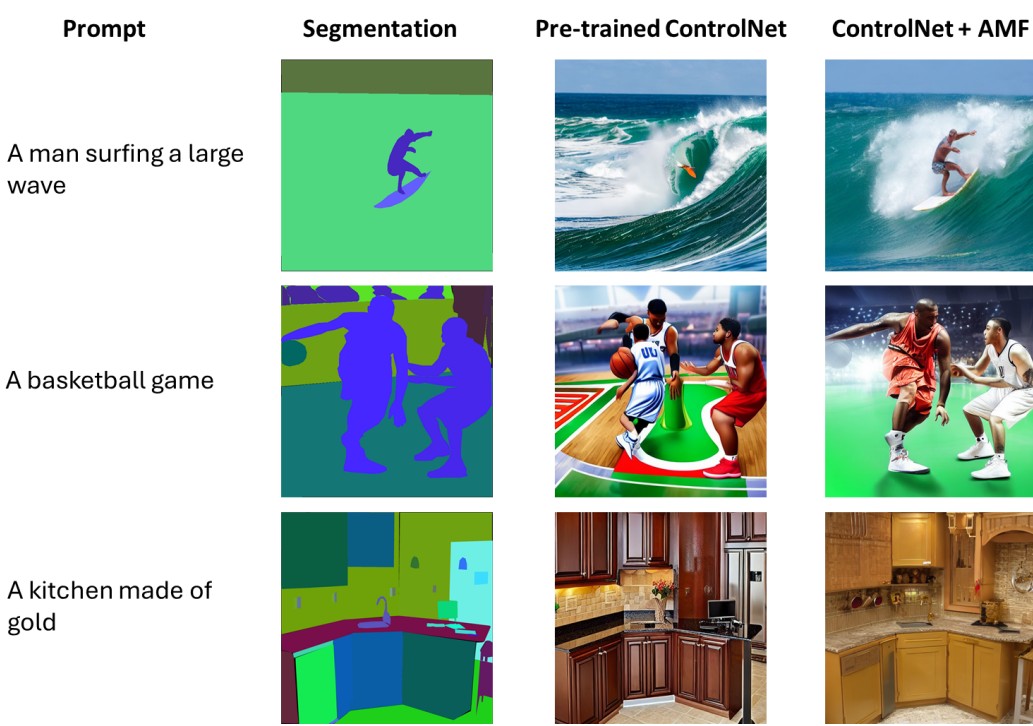

Figure 13: **Qualitative results for segmentation-conditioned ControlNet.** Qualitative results for pre-trained segmentation-conditioned ControlNet, without and with AMF. Both methods are initialized with the same seed. AMF leads to results that are more plausible and better align with the conditions, including intricate details (e.g. the base model fails to generate the faucet and the right stool in the third row, while our method generates both and better align with the prompt.)

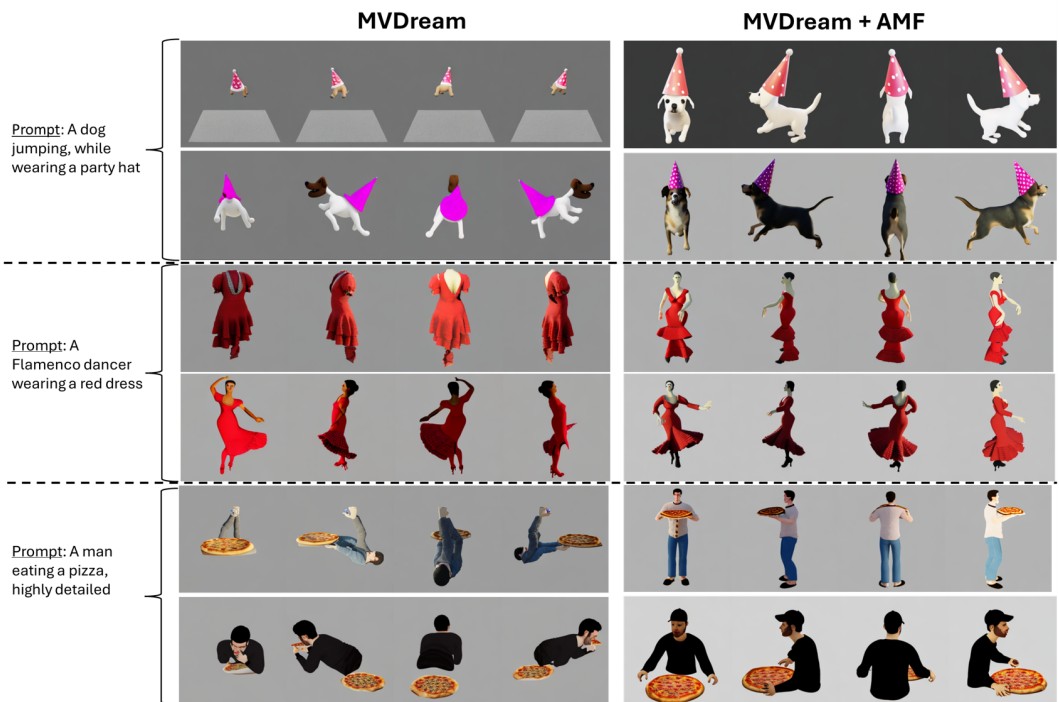

Figure 14: **Qualitative results for MVDream.** Qualitative results for MVDream, without and with AMF. Both methods are initialized with the same seed, and we provide results with two random seed per prompt. AMF leads to results that are more plausible and spatially consistent, while also better align with the conditions.

## 8.9 Data

**Dataset creation.** For each object, we render various views by randomly selecting camera parameters, an elevation $\theta \in [0, \pi$, an azimuth $\phi \in [0, 2\pi)$ and a radius $r \in [1.5, 2.2]$. We manually select 8 views for each object by removing images captured from close viewpoints and filtering camera viewpoints only from the upper half of the unit sphere, to avoid significant lighting effects that swing the results regardless of the output geometry.

**Dataset assets.** We provide a list of the objects picked for the challenging subset of GSO.

1. 3D_Dollhouse_Sink
2. 3D_Dollhouse_Swing
3. 3D_Dollhouse_TablePurple
4. adiZero_Slide_2_SC
5. Air_Hogs_Wind_Flyers_Set_Airplane_Red
6. BALANCING_CACTUS
7. Chelsea_lo_fl_rdheel_nQ0LPNF1oMw
8. CHICKEN_RACER
9. Circo_Fish_Toothbrush_Holder_14995988
10. COAST_GUARD_BOAT
11. CREATIVE_BLOCKS_35_MM
12. Dino_5
13. Down_To_Earth_Ceramic_Orchid_Pot _Asst_Blue
14. FIRE_ENGINE
15. Great_Dinos_Triceratops_Toy
16. Guardians_of_the_Galaxy_Galactic_Battlers _Rocket_Raccoon_Figure
17. Hilary
18. Imaginext_Castle_Ogre
19. My_First_Rolling_Lion
20. My_First_Wiggle_Crocodile
21. My_Little_Pony_Princess_Celestia
22. Nickelodeon_Teenage_Mutant_Ninja _Turtles_Leonardo
23. Nickelodeon_Teenage_Mutant_Ninja _Turtles_Michelangelo
24. Nintendo_Mario_Action_Figure
25. Nintendo_Yoshi_Action_Figure
26. Olive_Kids_Mermaids_Pack_n _Snack_Backpack
27. Ortho_Forward_Facing
28. Ortho_Forward_Facing_3Q6J2oKJD92
29. Ortho_Forward_Facing_QCaor9ImJ2G
30. Racoon
31. Razer_Kraken_Pro_headset_Full_size_Black
32. Remington_TStudio_Hair_Dryer
33. Rubbermaid_Large_Drainer
34. Schleich_African_Black_Rhino
35. Schleich_Hereford_Bull
36. Schleich_Lion_Action_Figure
37. Schleich_S_Bayala_Unicorn_70432
38. Smith_Hawken_Woven_BasketTray_Organizer _with_3_Compartments_95_x_9_x_13
39. Sonny_School_Bus
40. Sootheze_Cold_Therapy_Elephant
41. SORTING_TRAIN
42. Toysmith_Windem_Up_Flippin_Animals_Dog
43. Squirrel
44. Transformers_Age_of_Extinction_Mega_1Step _Bumblebee_Figure
45. TriStar_Products_PPC_Power_Pressure_Cooker _XL_in_Black
46. Vtech_Roll_Learn_Turtle
47. W_Lou_z0dkC78niiZ
48. Weisshai_Great_White_Shark
49. Whey_Protein_Vanilla
50. ZX700_mzGbdP3u6JB

