# OpenReview forum: "Zero-to-Hero: Enhancing Zero-Shot Novel View Synthesis via Attention Map Filtering"
_NeurIPS.cc/2024/Conference — NeurIPS 2024 poster_

### Official Review · Reviewer_hzMx · 2024-07-09

**Soundness:** 3
**Presentation:** 2
**Contribution:** 3
**Rating:** 5
**Confidence:** 5

**Summary:**

This paper proposes a test-time approach that adaptively modifies the attention map during inference to enhance the consistency and plausibility of novel view synthesis. Experimental results on GSO demonstrate improved performance with the proposed method, and the authors conducted ablation studies to assess the impact of different modules on performance.

**Strengths:**

1. The authors carefully analyze the challenges and issues of the current zero-1-to-3 method on novel view synthesis.
2. The idea proposed in the paper to filter attention maps akin to SGD is interesting as it can enhance the robustness of attention maps to a certain extent.

**Weaknesses:**

1. Since (Re-)Sampling increases the computational load, with the hyperparameter R determining the number of resampling iterations, it would be beneficial for the authors to conduct an ablation study on how this parameter affects both results and computational overhead.
2. The method proposed by the authors has shown limited improvement in results. They need to validate the effectiveness of their proposed approach on a larger and more diverse dataset.
3. When conducting the ablation study, the authors' selection of only 10 objects from GSO could introduce bias into the results. It would be preferable for them to include all objects to ensure a more comprehensive evaluation.
4. The authors' proposal of Mutual Self-Attention (MSA), which has been used in other papers such as Consistent123, Tune-A-Video, and MasaCtrl, cannot be considered a unique contribution in their paper.
5. In the ablation study (Table 2), Mutual Self-Attention (MSA) shows the largest improvement, with a greater increase in PSNR (MSA 17.82 vs AMF 17.58, last two rows). This suggests that the method's improvement primarily stems from MSA rather than the core contribution point, AMF.

**Questions:**

The paper includes several hyperparameters such as the number of resampling iterations (R) and the alpha for cross-step aggregation. How did the authors set these hyperparameters, and how significantly do they impact the final results?

**Limitations:**

The method proposed by the authors seems intriguing, but the actual improvement in results is limited. Moreover, the largest gain comes from a commonly used trick for enhancing viewpoint consistency, Mutual Self-Attention (MSA).

---

> ### Author Rebuttal · Authors · 2024-08-06
>
> We sincerely thank the reviewer for insightful feedback and comments.
>
> **1. Setting the Hyperparameters R and alpha.**
>
> All hyperparameters were tuned based on a small random set of objects. Due to limited computational resources, the tuning process was not exhaustive. We found that the method's performance is not highly sensitive to these parameters.
>
> - **Cross-step weight (alpha)**: We experimented with values ranging from 0.1 to 0.9 in increments of 0.1. Alpha determines the weight assigned to previous predictions in the cross-step aggregation. In Zero123-XL, early predictions were generally reliable, so a larger weight yielded better results. We maintained the same parameters for Zero-1-to-3. In new experiments conducted for the rebuttal, we applied our method to additional models such as ControlNet and MVDream. In both cases, a smaller weight generally produced better outcomes.
>
> - **Resampling iterations (R)**: We observed that the model's performance is relatively insensitive to the choice of R, with values between 4 and 8 yielding similar results. While some objects benefited from larger values (`~10`), the overall improvement was minimal. Additionally, as shown in Figure 8 in the paper's appendix, large values of R (`~15-20`) can reduce diversity. For additional experiments with models like ControlNet and MVDream, we fixed R at 5 and did not test other values.
>
> **2. Additional Computational Cost of Resampling.**
>
> Resampling iterations are computationally equivalent to adding more denoising steps, as both linearly increase the number of function evaluations (NFE). In the paper, we addressed this by counting the NFE and keeping it comparable to the base model to ensure a fair comparison. Our chosen value for R, and the specific timesteps where we applied it, resulted in mapping 26 denoising time steps to a total of 66 NFE. For further details, please refer to our response #1 to reviewer UpfM, where we discuss the individual computational overhead of each module.
>
>
> **3. The Extent of Quantitative Improvement.** Please refer to General Comment #4 for a detailed discussion of the quantitative improvements.
>
> **4. Evaluating on Additional Dataset.**
>
> We evaluated our approach on the RTMV benchmark [1], an out-of-distribution dataset consisting of 3D scenes, each contains 20 random objects. Here, we present the results for Zero123-XL, using the same hyper-parameters reported in the paper. Our evaluation shows improvements across all metrics compared to baselines. GSO and RTMV are the common datasets used for evaluation in Zero1-to-3 and its follow-ups.
>
> | |T|NFE|PSNR|SSIM|LPIPS|IoU|
> |-|-|-|-|-|-|-|
> |Base|25|25|10.51|0.589|0.396|71.5%|
> |Base|50|50|10.54|0.589|0.393|71.9%|
> |Base|100|100|10.53|0.588|0.392|72%|
> |Ours|26|66|11.18|0.619|0.372|73%|
>
> **5. Including all objects in the ablation study.**
>
> Thank you for your suggestion. In response, we have conducted the ablation study across all test objects, focusing on Zero123-XL. The new results corroborate the trends observed in our initial submission. We will ensure that the comprehensive evaluation is included in the camera-ready version.
>
> |Hourglass|Resample|AMF|MSA|PSNR|SSIM|LPIPS|IoU|
> |-|-|-|-|-|-|-|-|
> |-|-|-|-|17.27|0.854|0.162|76.4%|
> |+|-|-|-|17.44|0.855|0.161|76.6%|
> |+|+|-|-|17.7|0.857|0.160|77%|
> |+|+|+|-|17.92|0.858|0.157|77.8%|
> |+|+|-|+|18.25|0.862|0.155|77.6%|
> |+|+|+|+|18.35|0.864|0.153|78.3%|
>
> **6. The usage of Mutual Self-Attention in Zero-to-Hero.**
>
> We acknowledge that the formulation of MSA was not introduced in our work. We have cited prior works such as MasaCtrl in our paper, and we will include the additional references suggested by the reviewer. However, we wish to highlight the distinctions between previous works and our unique application of this technique.
> To our knowledge, most prior works utilizing MSA fall into two categories:
>
> * Training-free usage of MSA (e.g., MasaCtrl): In these works, MSA is typically applied after the general structure of the target is formed to transfer appearance details from the input to the target. In this scenario, MSA does not contribute to the initial structure formation of the target.
> * Training or fine-tuning models with MSA layers (e.g., Consistent123, Tune-A-Video): These works incorporate MSA within the training or fine-tuning process.
>
> Our approach is distinct as we employ MSA in a training-free manner, but crucially, we apply it from the beginning of the denoising process until the target structure stabilizes—a phase we term "Early-Stage Shape Guidance" in our paper. This early application of MSA leads to more stable results and often prevents the model from generating out-of-distribution outputs (like the leftmost and rightmost chairs in Figure 1 in the paper).
>
> To illustrate the impact of early-stage MSA on structure, we conducted a simple experiment using Zero123-XL with 50 DDIM steps. We measured the effect of activating MSA at different stages of denoising on Intersection over Union (IoU) metric, noting that image quality metrics improved similarly. Our approach demonstrated a significant improvement in the structural integrity of the image. We believe that applying MSA in the later stages of the denoising process introduces a bias towards the input images, which can disrupt the shape and appearance.
>
> |Method|Timesteps where MSA is applied|IoU|
> |-|-|-|
> |No MSA|-|76.4%|
> |All the way|1000 to 0|76.6%|
> |Start after the structure is initially formed|800 to 0|76.5%|
> |Ours (from the beginning and terminate early)|1000 to 600|77.6%|
>
> **7. The Contribution of MSA vs. AMF.**
>
> MSA and AMF address different artifacts and complement each other. MSA's greater impact on metrics does not diminish AMF's value. The combined methods show consistent improvements. For more details, please see General Comment #1 and Figure 1 (right) in the supplementary materials.
>
> [1] RTMV: A Ray-Traced Multi-View Synthetic Dataset for Novel View Synthesis.

---

> > ### Comment · Area_Chair_XFWB · 2024-08-11
> >
> > Dear Reviewer,
> >
> > Please take a moment to read the rebuttal if you haven't already, and assess the new information provided by the authors. Then please provide your updated comments.
> >
> > Thanks,
> > AC

---

> > ### Comment · Reviewer_hzMx · 2024-08-12
> >
> > Thank you to the authors for the detailed response. The rebuttal addressed most of my concerns, and I am willing to raise the score.

---

> > > ### Author Response · Authors · 2024-08-12
> > >
> > > We thank the reviewer for their thoughtful feedback and for taking the time to review our rebuttal and engage with us. Your comments are invaluable for us. We are pleased we could address your concerns and appreciate that you revised your evaluation accordingly.
> > >
> > > We are happy to address any further concerns.

---

### Official Review · Reviewer_3GQx · 2024-07-12

**Soundness:** 2
**Presentation:** 3
**Contribution:** 3
**Rating:** 5
**Confidence:** 3

**Summary:**

This paper proposes a novel approach to generate realistic images from arbitrary views based on a single source image. The authors introduce a test-time method that enhances view synthesis by manipulating attention maps during the denoising process of diffusion models. This process improves geometric consistency and realism without requiring retraining or significant computational resources. The key contributions include an attention map filtering mechanism, a modification of the self-attention mechanism to integrate source view information, and a specialized sampling schedule. Experimental results show substantial improvements in fidelity and consistency, demonstrating the effectiveness of the proposed method.

**Strengths:**

1. The paper introduces a unique test-time method for enhancing view synthesis, specifically manipulating attention maps during the denoising process, which is innovative and effective.
2. The paper is generally well-written and easy to follow.

**Weaknesses:**

1. Lack of qualitive results in the experimental part and even in Appendix I cannot find more comparison.
2. Only compared and equipped with zero123 and zero123-XL is not enough also.

**Questions:**

1. Can Zero-to-Hero be equipped with other multi-view diffusion models, e.g,,MVDream, Wonder3D?
2. The quantitive metrics is calculated in which views?
3. Will Zero-to-Hero hurt the diversity of Zero123 model with relatively consistent outputs among different seeds?

**Limitations:**

Please refers to W and Q. I will consider to raise my score if the author could address my concerns and provide more convinced results.

---

> ### Author Rebuttal · Authors · 2024-08-06
>
> We sincerely thank the reviewer for insightful feedback and comments.
>
> **1. Additional Qualitative Results.**
>
> We kindly refer the reviewer to General Comment #1 and the supplementary material, where we have included further qualitative results.
>
> **2. Applicability of Zero-to-Hero in Multiview Diffusion.**
>
> Thank you for raising this critical point. While Zero-to-Hero was initially designed for single-view generative models, our Attention Map Filtering can indeed be extended to other diffusion models, including multiview synthesis. Based on your suggestion and that of reviewer qQr5, we have implemented the attention map filtering mechanism for MVDream and two pre-trained ControlNet models, observing significant improvements. Please refer to General Comment #2 for more details on these implementations.
>
>
> **3. Quantitative Evaluation — Additional Details.**
>
> For our quantitative evaluation, we rendered 8 random views of each object and used each view as a source. For each source view, we generated the remaining 7 views, resulting in a total of 56 generation tasks per object (each defined by a unique source-target pair). Each target view was generated 3 times with different seeds, yielding a total of 168 images per object. We then averaged the scores across all views and objects to ensure a comprehensive evaluation.
>
>
> **4. Effect of Zero-to-Hero on Generation Diversity.**
>
> In general, excessive use of attention map filtering might reduce generation diversity. We have analyzed this in the appendix of our paper (Figures 8 and 9). However, we find that responsible usage usually preserves diversity. For instance, in Figure 1 of the main paper, the generated chairs display diverse back sides, and the turtles in the third row further demonstrate the model's ability to produce varied results. The balance between diversity and fidelity is complex and warrants further study. Often, the "diversity" observed in the base model includes artifacts and deviations from the real-world distribution, as illustrated in Figure 1 of the main paper and Figure 1(left) in the supplementary material.

---

> ### Comment · Reviewer_3GQx · 2024-08-11
>
> Thank you for your comprehensive rebuttal. The majority of my concerns have been addressed, particularly regarding the details of the Quantitative Evaluation, the applicability of Zero-to-Hero in Multiview Diffusion, and the additional qualitative results. However, I remain not entirely convinced about the purported balance between "diversity" and "fidelity". The attention map filtering is applied to a multi-view diffusion model, whose diversity should be largely preserved to maintain its non-deterministic nature as a diffusion model. Considering all these factors, I have decided to revise my score from borderline reject to borderline accept. I intend to consult other reviewers' discussions before making a final decision and I am more than happy  to discuss with other reviewers and  AC, SAC and PC about the diversity concern.

---

> > ### Author Response · Authors · 2024-08-12
> >
> > We thank the reviewer for their thoughtful comments and for taking the time to review our rebuttal, your feedback is invaluable to us. We are glad we could address your major concerns and appreciate your revised evaluation accordingly.
> >
> > Regarding your concern about generation diversity, we would like to offer some additional clarifications.
> >
> > While excessive resampling might reduce diversity (e.g. using large values of R), in practice we find that a moderate application of AMF is sufficient for improving fidelity while preserving diversity. Note that we only apply our filtering mechanism in the earlier steps of the denoising process, using a small value of R (5). **This enables our method to maintain diversity effectively.**
> >
> > When evaluating diversity against base models, we observed that some results of the base models were highly implausible, as can be seen in Figure 1 in the paper. For example, Zero-1-to-3 produces out-of-distribution results, or results that are not aligned with either the input image or the target pose, leading to seemingly larger variance. However, this "diversity" is largely due to misalignment and artifacts, which comes at the expense of fidelity.
> >
> > In our case, our model continues to generate diverse results that are both plausible and better aligned with the conditions (e.g. the various chair backs and turtles in Figure 1 of the main paper).
> >
> > **Future work.** Regarding your point about preserving diversity in multiview-diffusion, we agree that it is very important. In future work we plan to explore the effect of AMF in multiview-diffusion in depth and per your suggestion include a thorough analysis of diversity vs. fidelity. In particular, MVDream is conditioned on text rather than an image, making the desired solution space inherently more diverse which should be accounted for in such an investigation. That being said, in the preliminary results of MVDream (and also in the ControlNet models) that we ran for the rebuttal, we observed that the diversity is well-preserved.  For example, for the prompt “A man eating a pizza, highly detailed”, our method produced results that vary significantly in terms of viewpoint and appearance. This trend persisted across various prompts and seeds which we could not include in the rebuttal due to space limits.
> >
> > We thank the reviewer again for engaging with us, and are happy to address any further concern.

---

### Official Review · Reviewer_qQr5 · 2024-07-13

**Soundness:** 3
**Presentation:** 2
**Contribution:** 3
**Rating:** 6
**Confidence:** 5

**Summary:**

This paper proposes attention map filtering to enhance the novel view synthesis performance of Zero-1-to-3. The method is composed of several changes to the original sampling method, but the main contribution is an aggregation of attention map strategy by resampling the same denoising step multiple times. The paper draws an analogy to SGD to argue that resampling resembles the batch training in SGD, enhances the stability of the sampling process, thus improving the performance. Controlled experiments show that the proposed technique is effective in improving the quality of the generation

**Strengths:**

- The paper is well-written. Related works include sufficient references. The method is clear and well-structured. The figures are also well-made and clear. I had no trouble understanding the method and the results presented in the paper.

- The method is simple yet seems to be effective. The technique can be plug-and-play to apply to diffusion-based NVS methods such as Zero123.

- The analysis is great. The authors perform interesting analysis on attention maps of the denoising layers, the decreased diversity in generation, and limitations. I think these analysis helps a lot in building intuition behind the method and can be useful for future works to understand the nature of the problem.

- The experiments are controlled. Though the authors didn't compare against the various latest 3D generative models, they focus on evaluating the effectiveness of the proposed technique by running controlled studies. At a time when there are so many papers coming out every day, controlled and scientific experiments are valuable.

**Weaknesses:**

- Following the analogy introduced by the authors between diffusion and SGD, it seems unavoidable that aggregating attention maps across multiple sampling steps will compromise the model on generation diversity. As we can imagine, optimization with a larger batch size is more stable. So, even though the authors have proposed techniques to mitigate the problem of lack of diversity, I believe it is a more fundamental limitation.

- Since the techniques proposed seem general to all diffusion models, what prevents one from applying the technique to all diffusion generation tasks? Given the vast amount of literature and problems being solved by diffusion models, the potential impact could be significant. Are there any specific constraints such that the method is only applicable to the problem of novel view synthesis?

- Does the method lead to lack of visual details in the generated views? I noticed that the eyes of the generated chickens are missing in the last row of figure 1. Is this a general problem of the proposed techniques?

- It's a little unfair to claim that the proposed method can maintain the same computational efficiency. The proposed techniques to accelerate the diffusion process is not exclusive to the proposed techniques and can be applied to the base model as well.

**Questions:**

Please see the weakness section for questions.

**Limitations:**

The paper has discussed the limitations.

---

> ### Author Rebuttal · Authors · 2024-08-06
>
> We sincerely thank the reviewer for insightful feedback and comments.
>
> **1. Effect of AMF on Generation Diversity.**
>
> We agree that excessive use of Attention Filtering (R>>1) may reduce generation diversity, as analyzed in the appendix. We will incorporate these insights into the limitations section. The balance between diversity and fidelity is a complex topic that warrants further study. Our findings indicate that "diversity" in the base model often includes artifacts and deviations from the real-world distribution, as illustrated in Figure 1 of the main paper and Figure 1 (left) of the rebuttal Supplementary Material. Responsible use of Attention Map Filtering usually helps preserving diversity while preventing results from deviating from the real distribution.
>
> **2. Generality and applicability to other diffusion models.**
>
> Thank you for highlighting this important point. Although our work focuses on novel view synthesis, we agree that the AMF module is potentially broadly applicable. This was indicated in the future work section. Inspired by your comment and by the request of reviewer 3GQx, we have extended Attention Map Filtering to other models beyond the task of novel view synthesis. Remarkably, we achieved noticeable improvements out-of-the-box in all cases. Please refer to General Comment #2 for more details.
>
> **3. Visual details in generated views.**
>
> We have not observed a general trend of lacking visual details in the generated views. Figures 1 in both the main paper and the supplementary materials demonstrate examples with fine details, such as the Android's antennas and the squirrel's eyes. The loss of the chicken's eye in one example may be due to resampling issues. While our pipeline usually mitigates the oversmoothing effect of vanilla resampling, it is not infallible.
>
> **4. Computational efficiency.**
>
> We acknowledge the reviewer's point regarding computational efficiency. Our primary goals were to improve quality and consistency while keeping generation times competitive to ensure applicability. We will clarify our claims in the paper and are open to further adjustments based on the reviewer's suggestions. An analysis of the computational cost of each module is provided in the response to reviewer UpfM.

---

### Official Review · Reviewer_UpfM · 2024-07-24

**Soundness:** 3
**Presentation:** 3
**Contribution:** 3
**Rating:** 6
**Confidence:** 4

**Summary:**

This paper experimentally analyzes which parts are important and responsible for generation artifacts.
To solve it, this paper propose an attention map filtering process.
This process share the similar idea with SGD, which reduces the error of the generation process by repeated sampling.
This paper also propose some other things to enhance the results, including identity view information injection and a specialized sampling schedule.
The whole pipeline is training-free, so it’s very easy to apply to the pre-trained diffusion model.

**Strengths:**

This paper propose a method to improve the  novel-view diffusion model without external model and training, it is very easy to use and can be applied to different novel-view diffusion models easily.
The motivation of this article is clear, and the process and details of thinking and discovery is demonstrated.

**Weaknesses:**

* The model has no training cost, but there is a lack of thorough analysis of the additional inference cost.
* Qualitative results are limited. Given that the quantitative results do not show much improvement, more qualitative comparisons would be better.

**Questions:**

* Ablation studies of zero123 and zero123-XL showed different trends. MSA even impaired the results, but MSA with AMD improved the results. What is the reason do you think?
* Form the quantitative results, the improvement brought by MSA is significantly greater than that of AMF. Are there any qualitative results on MSA and AMF?
* How does this method affect the speed of the original diffusion model? Does the choice of R affect the model results?
* Fig 3 and 7 seem to share some of the same samples, maybe we can show more cases and perspectives?

**Limitations:**

This method is limited by the generative capabilities of the pre-trained model, if the results wrong too much, it does not have the ability to correct it.

---

> ### Author Rebuttal · Authors · 2024-08-06
>
> We sincerely thank the reviewer for their insightful feedback and comments.
>
> **1. Inference cost analysis**
>
> Our proposed modules add a computational overhead to the base model. In the paper, we addressed this by counting the overall number of function evaluations (NFE) and keeping it on par with the base model to ensure a fair comparison. As requested, we now discuss the individual computational overhead of each module. We will incorporate this analysis into the manuscript for completeness.
>
> (1) Resampling: Similar to the total number of denoising steps, T, the number of resampling iterations, R, linearly increases the NFE. Our chosen value for R and the timesteps at which we apply it resulted in mapping 26 denoising steps to a total of 66 NFE.
>
> (2) MSA: The main additional cost of MSA is the necessity to generate the input view in addition to the target views, meaning the effective number of generated samples is increased by one.
>
> (3) AMF: We implement attention map filtering in the upsampling part of the UNet and apply it during the early steps of the denoising process. We maintain two additional instances of each attention map: the attention map from the previous timestep (for cross-step updates) and the refined map from the current timestep (for in-step updates).
>
> We provide a table showing the running times (in seconds) of Zero-1-to-3 and Zero-to-Hero for the same NFE (66). Overall, we manage to provide competitive running times. If both MSA and AMF are active (requiring the generation of the input), the running time is increased by approximately 1-1.5 seconds. If only AMF is active, the overhead is much smaller, averaging around 0.5 seconds.
>
>
>
> |Samples Num|Zero123|Ours|
> |-|-|-|
> |1|2.2|N.A|
> |2|2.9|3.1|
> |3|3.5|3.9|
> |4|4.3|4.9|
>
> **2. Additional qualitative results.**
>
> Please refer to Figure 1 in the supplementary material for additional qualitative comparisons of view synthesis by our Zero-to-Hero and the baseline Zero-123-XL. As can be seen, the improvement demonstrated by our method is consistent across objects and seeds. Further qualitative results demonstrating our method on other base models are discussed in General Comment #1.
>
> **3. The extent of quantitative improvement.**
>
> We have provided a clarification in General Comment #4.
>
> **4. Qualitative demonstration of MSA and AMF.**
>
> We appreciate this suggestion. Indeed, the difference in quantitative improvement between these two modules does not faithfully reflect their effect. We thus chose to share the response with all reviewers in General Comment #1 and Figure 1 (right) in the supplementary material.
>
> **5. The effect of MSA and AMF on zero123 and zero123-XL.**
>
> This is a good observation. Firstly, regarding the different effects of MSA and AMF, please refer to General Comment #3. Here, we address the reviewer's inquiry regarding the *difference in the modules' effects on both base models*. Our MSA (MSA with early termination) generally improves performance in both base models. As pointed out by the reviewer, in Zero-1-to-3, MSA did not improve PSNR and IoU while boosting SSIM and LPIPS. In Zero123-XL, the improvement was consistent across all metrics. A potential explanation for this difference may lie in the inherent bias of MSA towards the input view. Roughly speaking, MSA copies source details to improve generation consistency. Thus, too much of it might impair the results as it would lead to a bias towards the appearance and pose of the input image. To control this effect, we introduced early stopping. The termination step in our work was chosen based on Zero123-XL (where MSA consistently improved all metrics). While we found that the same parameters generalized well for the overall performance of Zero-1-to-3 without further tuning (the combined MSA and AMF improved all metrics and addressed the same issues as Zero123-XL in visual results), the ablation reveals that the parameters may not be optimal for the MSA effect alone in Zero-1-to-3.
>
> **6. Choice of resampling steps R.**
>
> We found that the model is not very sensitive to the choice of R, and values within the range of 4-8 provide similar results. While some objects benefited from larger values (`~10`), the overall improvement was minimal. Additionally, we found (as shown in Figure 8 in the appendix of the paper) that large values of R (`~15-20`) can limit diversity.

---

> > ### Comment · Reviewer_UpfM · 2024-08-13
> >
> > Thank you for the response. The rebuttal addressed most of my concerns, and I am willing to keep my score.

---

> > > ### Author Response · Authors · 2024-08-13
> > >
> > > We thank the reviewer for their thoughtful comments and for taking the time to review our rebuttal and engage with us. Your feedback is invaluable for us.
> > >
> > > We are pleased we could address your concerns, and are happy to address any further concerns.

---

### Author Rebuttal · Authors · 2024-08-06

We sincerely appreciate the reviewers and ACs for their efforts in reviewing our work.
We are encouraged by the recognition of our work's innovative perspective of attention map filtering as analogous to optimization process (hzMx, 3GQx), applicability and effectiveness (UpfM, qQr5). We are pleased that the thoroughness of our analyses was appreciated (UpfM, hzMx, qQr5).  Below, we address the concerns raised by the reviewers.

**1. Additional results**

We provide additional qualitative and quantitative results in the Supp. These results reinforce the effectiveness of Zero-to-Hero and demonstrate its generalization to additional datasets and models. (1) Figure 1 (left): Additional view synthesis results of Zero-to-Hero. Figure 1 (right): results that exemplify the individual contributions of the proposed MSA and AMF. Figure (2): Our AMF applied to pose- and segmentation-conditioned generation pre-trained with ControlNet. Figure (3) Our AMF applied to Multiview synthesis model MVDream. (4) Evaluation on the RTMV, a dataset of 3D scenes. Please refer to our response to reviewer hzMx for further details.

**2. Zero-to-Hero generalization beyond NVS**

Although our work addresses the core limitations of single view synthesis models, the condition enforcing effect of our proposed modules are more general. As mentioned in the conclusions, we intend to leave the in-depth exploration of other applications to future work. However, as the generality of our method seemed to draw much interest by the reviewers we have conducted several preliminary experiments which demonstrate promising results. Remarkably, the integration of our proposed method into other base models was straightforward, demonstrating its applicability and simplicity.
(1) Pose- and Segmentation-conditioned image generation. A brief study of ControlNet models demonstrated that they suffer from similar limitations as zero123 and its follow ups. Namely, lack of condition enforcement and frequent appearance of visual artifacts. We implemented our proposed AMF module for two pre-trained ControlNet models and found that it robustly mitigates artifacts across various prompts and seeds. Please note that MSA is not immediately applicable for these models and thus was not used.
(2) Multi-view synthesis. We integrated AMF into MVDream, a text-to-multiview model, and found that it helps to mitigate the same issues as in the single view case. Similarly, MSA was not implemented.

**3. The Contribution of MSA vs. AMF**

Reviewers UpfM and hzMx, while the image quality metrics show a larger improvement with MSA, these numbers do not tell the whole story. AMF and MSA address different artifacts in the generation process and complement each other. The fact that MSA contributes more to the metrics does not take away from AMF's individual contribution. This complementary effect is demonstrated by the consistent improvement observed when both methods are used in tandem compared to using each individually.

- **MSA** (Mutual Self-Attention) transfers information from the input view to the target, assuming similar appearance and textures. This method is particularly effective when the input and target views are relatively close. As an example, Zero123 sometimes generates regions with plain black or random textures in the target views which MSA mitigates well, as shown in the first two rows of Figure 1 (right) in the Supp. We note that image quality metrics are more sensitive to this improvement and thus show more significant gains than when improvement in shape is achieved. However, MSA alone is usually insufficient for refining the pose or structure of the target. Its inherent bias towards the input shape and appearance may harm the results, especially when the change of viewpoint is significant. Our analysis therefore led us to utilizing MSA only during the early steps of the denoising process, named Early-Stage Shape Guidance in our paper.

- **AMF** (Attention Map Filtering), on the other hand, excels when the change in viewpoint is larger. While it may not always improve color and textures, AMF leads the model to produce more probable results that align better with the real distribution. We observed that most of the structure refinement is done by AMF. Unfortunately, none of the metrics measure plausibility, so this effect is not faithfully reflected by the evaluation metrics.

We have included a new figure demonstrating where MSA and AMF excel. The first two rows illustrate why MSA shows a larger improvement in image quality metrics, although it usually cannot fully resolve significant structural issues. Rows 3 and 4 show the structural improvement achieved with AMF. Finally, the last row shows a case where neither technique worked well enough on its own, but the combination did the work.

As further testimony to the significant effect of AMF, we refer the reviewers to general comment #2, where AMF is used to boost other generative models such as ControlNet and MVDream. These examples exhibit similar issues to Zero123 and demonstrate the role of AMF in mitigating them.

**4. The Extent of Quantitative Improvement**

Reviewers UpfM and hzMx raised a concern regarding the extent of improvement in the quantitative results reported in Table 1 of the main paper. While the absolute improvement may not seem large, it is important to put it in context to appreciate its significance. Zero123-XL improved upon Zero123 using the same base model by using 10x more data, achieving gains of [0.45, 0.003, -0.01, 2.9%] in PSNR, SSIM, LPIPS, and IoU, respectively. Our method achieved comparable gains [0.37, 0.008, -0.01, 1.7%] with no additional data and no further training. Remarkably, our method demonstrates similar and slightly larger gains when applied to Zero123-XL [0.63, 0.1, -0.01, 1.9%], a boost in performance that couldn't be achieved with merely more data. Also, the ratio of improvement is larger compared to other training-free methods (e.g. ViVid123).

---

### Decision · Program_Chairs · 2024-09-25

**Decision:**

Accept (poster)

**Comment:**

The reviewers acknowledge the technical contribution of proposing a training-free method and its effectiveness, along with a clear presentation. The rebuttal was successful, resulting in an increased average score, with previous negative scores now becoming supportive. After careful discussion and consideration, we are pleased to inform you that your paper has been accepted. However, the final version will require revisions to reflect the important discussions presented in the rebuttal. For example, please discuss the diversity issue, clarify the distinction from the previous MSA module, and include additional experimental results.